# Ecoresorbable chipless temperature-responsive tag made from biodegradable materials for sustainable IoT

James Bourely [1] ✉, Nicolas Fumeaux[1], Xavier Aeby[2], Jaemin Kim[1], Gilberto Siqueira [2], Christian Beyer [3], David Schmid [3], Oleksandr Vorobyov[3], Gustav Nyström[2,4] ✉ & Danick Briand [1] ✉

Temperature monitoring within the cold chain, essential for safety of perishable products, typically employs devices such as battery-powered data loggers and radio-frequency identification tags. Such devices include non-eco-friendly components, posing challenges for their safe disposal and recycling. This study demonstrates the fabrication of a fully ecoresorbable, chipless, and wireless temperature-responsive tag, designed to irreversibly track temperature changes through a permanent shift in resonance frequency. The tag is printed on a customized moisture-resistant poly(β-hydroxybutyrate)-cellulose composite substrate. An RLC circuit made of printed zinc metallic traces, encapsulated with beeswax to prevent oxidation, enables seamless wireless operation. The tag utilizes bio-based phase-changing materials such as frozen olive, jojoba, and coconut oils to induce irreversible resonance frequency shifts of more than 30 MHz at respective melting points of 8 °C, 15 °C, and 25 °C. A cellulose capillary element efficiently absorbs the melted oil, enabling reliable operation at inclinations from 0° to 90°. At the end of its service life, the device can undergo disintegration in a compost environment within 9 weeks. This work demonstrates a sustainable chipless technology from material selection and manufacturing processes to end-of-life disposal as an advanced thermal indicator solution for cold chain temperature-excursion detection.

Ensuring accurate temperature monitoring in the cold chain is crucial for maintaining the quality and safety of perishable goods[1–3]. As global electronic waste (e-waste) increases at a rapid pace and poses a major challenge, the end-of-life disposal of the current battery-powered data loggers applied to cold chain monitoring and the expanding deployment of radio-frequency identification (RFID) tags contribute to this environmental problem[4–7]. These systems, often single-use, contain toxic materials integrated with silicon electronic components, which complicate or even prevent their harmless disposal or recycling[8–10].

Printed electronics can lead to a more sustainable production by additive manufacturing of low-complexity electronic systems, such as RFID tags, notably by reducing material usage[11–13]. These tags still rely on silicon chips, with some concepts being recently proposed for their detachment, envisioning a more efficient e-waste management, by enabling the proper recuperation or recycling of the different

[1]Ecole Polytechnique Fédérale de Lausanne (EPFL), Soft Transducers Laboratory (LMTS), Rue de la Maladière 71b, Neuchâtel, Switzerland. [2]Swiss Federal Laboratories for Materials Science and Technology (Empa), Cellulose & Wood Materials Laboratory, Überlandstrasse 129, Dübendorf, Switzerland. [3]CSEM SA, Rue Jaquet-Droz 1, Neuchâtel, Switzerland. [4]ETH Zürich, Department of Health Science and Technology, Zürich, Switzerland. ✉e-mail: james.bourely@epfl.ch; gustav.nystroem@empa.ch; danick.briand@epfl.ch

components[14]. From a sustainability standpoint, efforts have been focused on switching from conventional plastics to renewable and recyclable substrates by using paper[15]. However, some work remains to be performed on the transducing materials, with conductive silver or copper inks being mainly used to form, by printing, the wireless tags[16–21].

Towards achieving improved sustainability for such systems, there is a growing trend in implementing eco-friendly biodegradable materials, that can degrade naturally without releasing harmful by-products to further reduce the environmental impact[22–29]. When applied to printed electronics, the substrates are desired to exhibit a certain tolerance to temperature and limited roughness and hygroscopicity, which is mainly not the case for the biopolymers or resins demonstrated so far, such as polylactic acid (PLA), chitosan, and shellac[30–32]. Cellulose substrates, known for their affordability and widespread availability, are particularly notable for their relatively high temperature tolerance (~ 230 °C) in comparison to many other biopolymers[33]. However, the dimensions of conventional cellulose fibers and paper-making methods of pressing and dewatering can make cellulose rough and porous, posing challenges when printing and for the stable operation and reliability over time of radio-frequency (RF) systems when exposed to humidity[33,34]. While finer cellulose materials, e.g., nanocellulose, may help to improve surface roughness, barrier coating or chemical modification, which mostly involve toxic materials such as alkene ketene dimer or alkenyl succinic anhydride, are typically needed to improve humidity stability[35–38]. Therefore, it is crucial to develop robust biodegradable substrates for printed electronics and RF applications. One promising approach involves compounding cellulose with hydrophobic biodegradable polymers, such as semicrystalline polycaprolactone with a low melting point of 60 °C[39]. Some efforts have also been dedicated to designing, by printing, highly electrically conductive degradable metallic films based on zinc suitable for wireless communication. The zinc bioresorbability and degradation mechanisms via electrochemical corrosion, forming biocompatible zinc ions and oxides, make it particularly attractive for transient electronic applications[40–46]. The deposition and sintering methods for zinc conductive traces are compatible with biopolymeric and cellulosic-based substrates[47,48]. However, silicon-based RFID tags made solely of degradable structural and functional materials have not yet been demonstrated.

Passive chipless RLC resonators are emerging as promising alternatives to conventional silicon-based RFID sensors, with most implementations relying on standard printed circuit board materials like copper on FR4 and recent progress involving silver printed on polyethylene terephthalate (PET) or paper[49–55]. These devices operate by modulating electromagnetic waves, eliminating the need for silicon integrated circuits and have been used for sensing and identification[56–58]. By exploiting phase-changing materials (PCMs), chipless thermal indicators to irreversibly track changes in temperature, operating in the GHz range, can be realized[58,59]. Covering the RLC tag locally with PCMs like paraffin or bio-based fatty acids, and their spreading once reaching their melting temperature result in a modification of the dielectric and RF properties of the tag, leading to an irreversible shift in its resonance frequency[60,61]. PCMs offer significant advantages due to their hydrophobic nature and ease of integration into chipless devices for the irreversible detection of temperature exceedance[62]. While incorporating bio-based PCM materials such as grapeseed or coconut oils, these chipless temperature-responsive devices do not rely on printing technology, utilizing either etched copper or transferred aluminum on non-biodegradable substrates[63,64]. In summary, a holistic approach for the realization of sustainable tags for comprehensive cold chain temperature-exceedance detection is missing.

In this communication, we leverage solely eco-friendly materials and mass-production compatible additive manufacturing processes to develop sustainable tags for cold chain monitoring. Our approach relies on printing biodegradable materials to produce a wireless PCM-based indicator to irreversibly track temperature. The tag employs a chipless architecture, eliminating the need for silicon electronic chips, using as a transducer an RLC electrical circuit covered by a natural oil that melts at a specific temperature. Zinc conductive ink is printed on a newly degradable substrate made of poly(β-hydroxybutyrate) (PHBH) composited with cellulose, with a beeswax film for oxidation protection. As PCMs, frozen olive, jojoba, and coconut oils integrated over the tag induce irreversible resonance frequency shifts of more than 30 MHz upon melting at 8 °C, 15 °C, and 25 °C, respectively. A cellulose capillary element is introduced, enabling the proper operation of the devices at angles from 0° to 90° by absorbing the melting oil. The device disintegrates in compost over a few weeks, highlighting the potential to avoid electronic waste after its service life. This successful wireless demonstration of irreversible monitoring of temperature at various thresholds using entirely biodegradable materials sets the ground for sustainable thermal indicator solutions applied to smart packaging.

## Results

### Eco-friendly PHBH-cellulose substrate

The development of robust biodegradable substrates for printing electronics and RF applications demands to achieve a low roughness, good thermal stability, low sensitivity to moisture, and eco-friendly disposal by being recycled or degraded using the established streams. Non-petroleum-based biopolymers with thermal stability above 50 °C are available, such as cellulose, starch, chitosan, polyhydroxyalkanoate (PHA), polylactic acid (PLA), polyglycolide (PGA) and polybutylene succinate (PBS), with only cellulose, starch, chitosan and PHA being home compostable. But cellulose, starch and chitosan are hygroscopic, and water intake could affect device performance and reliability due to mechanical stress or change in dielectric properties.

Our original approach to develop a biodegradable substrate having less sensitivity to moisture relies on compositing poly(3-hydroxybutyrate-co-3-hydroxyhexanoate (PHBH), which is hydrophobic, with cellulose. The latter has the benefits of being more tolerant to temperature, widely available at low cost and having a well-established recycling stream. PHBH is a bacterial synthetized biopolymer from the PHA family with a melting temperature of 130 °C[65]. Cellulose acts as a rigid filler and, by varying its content, allows tuning of the mechanical properties and the biodegradation rate of the substrate as it will be shown. The PHBH-cellulose composite is produced through melt compounding and reactive extrusion. The reaction happens in a twin-screw extruder, at 175 °C, with a range of cellulose fillers of 0 – 50 wt%. The limit of fiber volume content is determined by the geometry and alignment of the fibers in the polymer[66]. The production of PHBH with higher cellulose fiber concentrations is not achievable due to tearing of the composite after extrusion when fibers in random orientation exceed approximately 50 wt%. Choosing to blend fibers in the biopolymer allows for a reduction of PHBH material usage for the substrate, currently with low market availability and high cost due to its confidential manufacturing[65]. After the extrusion, the polymer is pelletized and hot pressed to form 200 µm-thick foils. The substrates produced are flexible and have a low surface roughness, as can be seen in Fig. 1a. Tensile tests performed on PHBH substrates with 0, 20 and 50 wt% cellulose loadings (supplementary Fig. S1) show that the more cellulose fibers, the higher the Young's modulus and the lower the elongation at break.

The roughness of the 200 µm-thick PHBH-50%cellulose composite is lower than that of a paper produced from nanofibrillated cellulose (nanopaper) and a commercially available cellulose substrate for printed electronics (PE paper). We compared the roughness using AFM for different scan areas with the resulting arithmetic roughness presented in Fig. 1b. The surface roughness for a scan area of 5 ×5 µm²

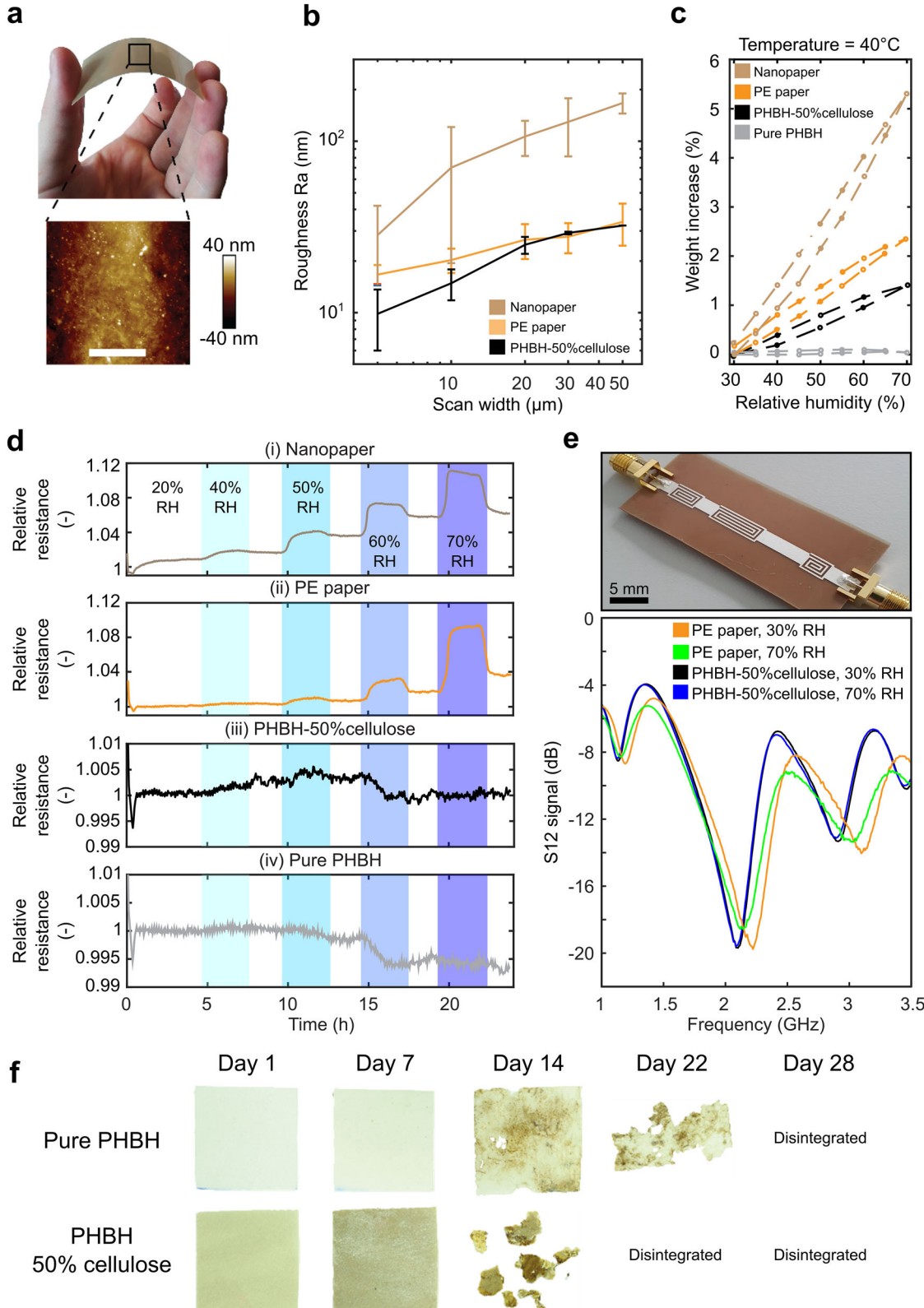

is 9.8 ± 3.8 nm, 16.7 ± 2.3 nm and 28.3 ± 13.6 nm, for the PHBH-50%cellulose, PE paper and nanopaper respectively ($n = 5$), increasing slightly to 32.2 nm, 33.9 ± 9.3 nm, and 130.7 ± 4.0 nm for a scan area of $50 \times 50\ \mu m^2$ ($n = 3$).

The mass variation of the 200 ± 15 μm-thick substrates as a function of the relative humidity (RH) was also lower for the PHBH-cellulose composite (Fig. 1c), with mass uptakes from 30% to 70% relative humidity at 40 °C being 5.3%, 2.3% and 1.4% for the nanopaper, PE paper and the PHBH-50%cellulose substrate, respectively. This shows that the developed PHBH-cellulose composite is less hygroscopic than the other cellulose-based materials considered and therefore more adapted for RF tag applications.

Cellulose content in the composite acts as nucleating agents, therefore the size of polymer crystals is expected to be smaller in the

**Fig. 1 | Properties of the biodegradable PHBH-cellulose substrate. a** Photograph of the 200 µm-thick PHBH-50 wt% cellulose composite, demonstrating its flexibility, with an inset showing its surface structure obtained by AFM. **b** Arithmetic average roughness as a function of the width of the scan area for 200 ± 15 µm-thick substrates of nanopaper, PHBH- 50%cellulose composite and the PE paper, error bars are from AFM. **c** Dynamic vapor sorption measurements, representing the relative mass variation of the 200 ± 15 µm-thick samples as a function of the relative humidity at 40 °C for nanopaper, PE paper, PHBH-50%cellulose composite and pure PHBH. **d** The relative resistance variations of the silver-printed traces on the nanopaper, PE paper, PHBH-50%cellulose and pure PHBH substrates when exposed to different levels of relative humidity from 30 to 70% at 25 °C. **e** Printed silver microstrip line on 200 µm PHBH-50%cellulose and the corresponding $S_{12}$ signal responses on PHBH-50%cellulose and 230 µm PE paper substrates at relative humidity of 30% and 70% RH and 25 °C. **f** Optical images over time of the pure PHBH and 50 wt% cellulose-loaded PHBH substrates placed in a custom-made compost (ISO 20200). The initial mass of the PHBH was 160.1 mg, and the mass of the composite was 173.0 mg, with both substrates having the same surface area. The composite was 3.8% thicker than the pure PHBH sample. Compost particles, sticking to the polymer matrix, visible as dark spots on the optical images, prevented the accurate measurement of the weight of the samples and therefore the quantification of the degradation rate.

50% cellulose composite resulting in a slightly lower melting temperature (129 °C) for the polymer blend compared to pure PHBH ($T_g = 131$ °C) as can be seen in the heat flow measurements where smaller crystallization and melting peaks are visible (Supplementary Fig. S2a). The trough observed at approximately 50 °C corresponds to the cold exothermic crystallization temperature of PHBH. This crystallization process arises from the alignment of polymer chains, facilitated by the material softening as it surpasses the glass transition temperature. Notably, the troughs and peaks in the thermal analysis are consistent with similar findings reported in other scientific studies on PHBH[67,68]. Cellulose being stable up to roughly 200 °C, the thermogravimetric analysis shows a temperature stability improvement of 20 °C for the PHBH-50%cellulose composite when compared to the pure PHBH. The cellulose particles help to stabilize the polymer at higher temperatures, and a residue of about 7% is observed for the PHBH-50%cellulose composite (Supplementary Fig. S2b). PHBH-50% cellulose with printed silver tracks can withstand usual sintering temperatures used for printed metallic inks, with bubbles starting to appear on the surface of the composite after exposure at temperatures higher than 140 °C for 30 min. Following annealing above 140 °C, the substrate loses most of its mechanical strength and becomes too soft to be properly manipulated, leading to the breaking of the silver resistors (Supplementary Fig. S3).

To evaluate the influence of the water vapor sorption properties of the composite on the electrical stability of printed metallic films, we measured the electrical conductivity of silver traces screen-printed on a 200 µm-thick PHBH-cellulose substrate when successively exposed to various step variations in relative humidity, from 30 to 70% RH, at constant temperature. The results and their comparison with pure PHBH, nanopaper and PE paper are shown in Fig. 1d. The PHBH-cellulose provided a much higher stability for the metallic traces with a negligible effect of the humidity on their electrical resistance, i.e., <0.5%, and this for the whole humidity range tested, as opposed to the nanopaper and PE paper cellulose substrates for which a drift was observed. The change of resistance as a function of the humidity is negligible from 30% to 50% relative humidity for the PE paper and slightly increases by 1–2% for the nanopaper, but reaches up to 9–10% at 70% RH for both, when compared to 30% RH.

The lower moisture absorption of the PHBH-cellulose substrate results in a more stable RF signal transmission with varying ambient relative humidity compared to the PE paper (Fig. 1e). Using a silver microstrip line printed on both the PHBH-cellulose and PE paper substrates, we investigated the effect of relative humidity at 25 °C on the RF S12 signal at ultra-high frequency from 1 GHz to 4 GHz[48]. Only a small change of 5 MHz in resonance frequency (< 0.3% variation in resonance frequency) for the printed resonators on PHBH-50% cellulose is observed for a change in relative humidity from 30% to 70% RH, which is twenty-two times smaller than the 110 MHz shift measured for the PE paper.

Finally, the 200 µm-thick PHBH-50% cellulose can be disintegrated in a lab-made compost environment following ISO 20200 for the disintegration of plastic materials in simulated composting conditions. The composite readily disintegrates in 3 weeks, faster than the pure PHBH, which decomposed after 4 weeks (Fig. 1f). The decomposition of the PHBH substrate was therefore facilitated when compounding with cellulose. This property could be used to tune the disintegration rate by adapting the cellulose content according to the end-of-life degradation scenario envisioned in practice[69].

Overall, the eco-friendly PHBH-50% cellulose composite offers a reduction in PHBH usage, minimizing material cost, while improving degradability and provides thermal and moisture-resistive properties adequate for printed electronics.

## Chipless tag design and test setup

This chipless temperature-responsive tag made exclusively of biodegradable materials is conceptualized to be attached to packaging in order to irreversibly monitor breaks in the cold chain (Fig. 2a). The device is shaped like a coin of 22 mm in diameter with the metallic RLC circuit printed on a PHBH-50%cellulose substrate 304 ± 57 µm-thick ($n = 16$). The thickness of the PHBH composite have been chosen to prevent undesired warping that could negatively impact the response of the tag when interfaced with a microwave reader antenna operating in the near-field. The RLC resonator consists of interdigitated electrodes (IDEs) as a capacitor confined by a conductive line shaped in the word "GREEN" as an inductor and resistors. The IDE design has been optimized for robustness towards the screen-printing process while minimizing the signal interference of various packaging materials onto which the tag could be attached[70]. The IDE is made of 4 pairs of interdigitated electrodes with a linewidth of 500 µm and a gap size of 300 µm for an overall footprint of $6 \times 8$ mm². The irreversible response mechanism is based on the phase transition of a frozen phase-change material (PCM) above its threshold temperature $T_{th}$. Upon melting, the spreading of the dielectric liquid over the IDEs area will result in an increase in the RF losses, thus leading to a reduction of the resonance frequency of the tag (Fig. 2b). An exploded view of the tag with the stack of the biodegradable materials involved and their thickness can be seen in Fig. 2c.

The cross-section of the devices is depicted in Fig. 2d for each step of the fabrication process with its corresponding 3D schematic. The zinc ink is screen printed over the 300 µm-thick substrate and sintered using a hybrid chemical and photonic flash sintering method shown to reach a very high electrical conductivity for a printed zinc layer[47,71]. The PHBH-cellulose polymer composite withstood the 10% acetic acid treatment used to dissolve the thin oxide layer covering the zinc particles. The same level of photonic energy (i.e., approximately 5 J/cm²) from the multi-pulse exposure could be delivered as reported for printed zinc on paper substrate, but distributed over one more pulse (i.e., 3 instead of 2) to lower the temperature reached at the surface of the PHBH composite[71]. As seen in supplementary Fig. S4 a substrate of 300 µm in thickness was preferred over 200 µm to prevent any warping of the tag due to the photonic sintering step. The RF tag, with its 100 µm-thick hydrophobic beeswax encapsulation added to protect the 15 µm-thick biodegradable zinc layer from oxidation, is visible in Fig. 2a. The pristine tag after encapsulation can be fabricated in a repeatable way with an average resonance frequency of 1.825 ± 0.029 GHz ($n = 7$). The pristine tag with and without beeswax

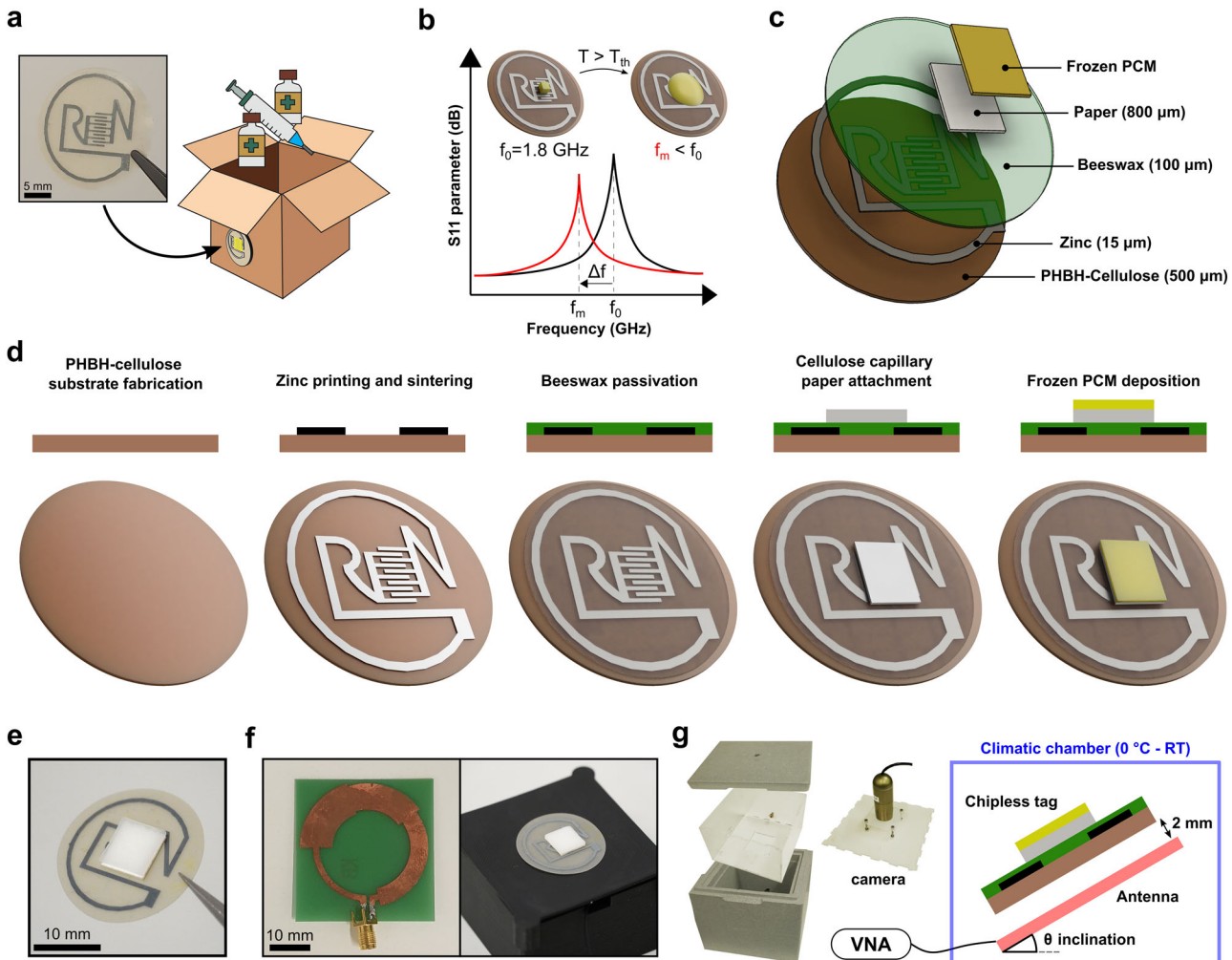

**Fig. 2 | Eco-friendly chipless temperature-responsive tag concept, fabrication and testing setup. a** Image of the chipless tag and schematic of the monitoring concept for smart packaging application. **b** Schematic of the operation principle based on a chipless tag with pristine and melted PCM resulting in a resonance shift towards lower frequencies. **c** Exploded view of the multi-layered biodegradable tag and material thicknesses. **d** Cross-section view and 3D representation of the manufacturing process flow. **e** Optical image of the completed chipless tag before use. **f** Optical images of the reader antenna and chipless tag with PCM positioned over the antenna with a 2 mm spacer. **g** Climatic setup with a cold chamber and a camera, and a schematic showing the tag placed over the reader antenna inside the chamber at an inclination angle θ.

was bent over various radii of curvature, as seen in supplementary Fig. S5, and can withstand a radius of curvature down to 10 mm with little impact (3 MHz shift) on the frequency of resonance after bending (Supplementary Fig. S6). Despite the tag not being intended to be flexible, it maintained a detectable resonance frequency after bending over 100 cycles at a radius of curvature of 15 mm, with an observed shift in resonance of 4 MHz (Supplementary Fig. S7). A 10 mm radius of curvature leads to failure after 100 cycles with a loss of the frequency of resonance and a shift in resonance of 15 MHz after 80 cycles. In both cases, the signal amplitude got reduced over time due to the deterioration of the zinc conductive trace and creasing of the PHBH substrate. Furthermore, after 1 year at ambient conditions, the frequency of resonance of the pristine encapsulated tag drifted by only 4 MHz (Supplementary Fig. S8). While the signal amplitude got reduced after 1 year, allegedly due to the slight oxidation of the zinc trace, the aged tag remained functional, and the accurate melting of a PCM for temperature exceedance was able to be detected. These results indicate that the beeswax-encapsulated tag can maintain radio-frequency functionality over an extended period, despite being composed of degradable materials. To allow wireless operation at different inclinations by absorbing the melted PCM through capillary effects, as will be

shown, the final version of the chipless thermal-responsive device also includes an extra 800 µm-thick cellulose absorbent fixed over the IDEs using an eco-friendly polyvinyl alcohol-based glue. Ultimately, olive, jojoba or coconut oils, melting respectively at 8, 15 and 25 °C and used as PCMs, are casted into molds to form frozen oil slabs placed over the paper well. Once fabricated, the devices are stored in a fridge, below the melting temperature of the PCM, until usage.

For RF wireless probing, the chipless tag with PCM is placed over a microwave near-field reader antenna operating between 1 GHz and 6 GHz, at a constant distance of 2 mm using a spacer (Fig. 2f). The custom-made climatic chamber setup, non-metallic to prevent RF interference, used to monitor the response of the tag as function of temperature is shown in Fig. 2g. Simulation of the tag and antenna gain, and 3D pattern indicate stable directional performance across the frequency range, making the antenna suitable for the chosen application (Supplementary Figs. S9, S10). The theoretical visualization of the tag and antenna shows that they remained coupled at 5 mm distance, with a decoupling occurring at 20 mm (Supplementary Fig. S11). The $S_{11}$ parameters of the antenna and the tag are measured using a vector network analyzer (VNA), and the reader antenna with the tag can be oriented at an inclination angle θ of 0° up to 90° to simulate the

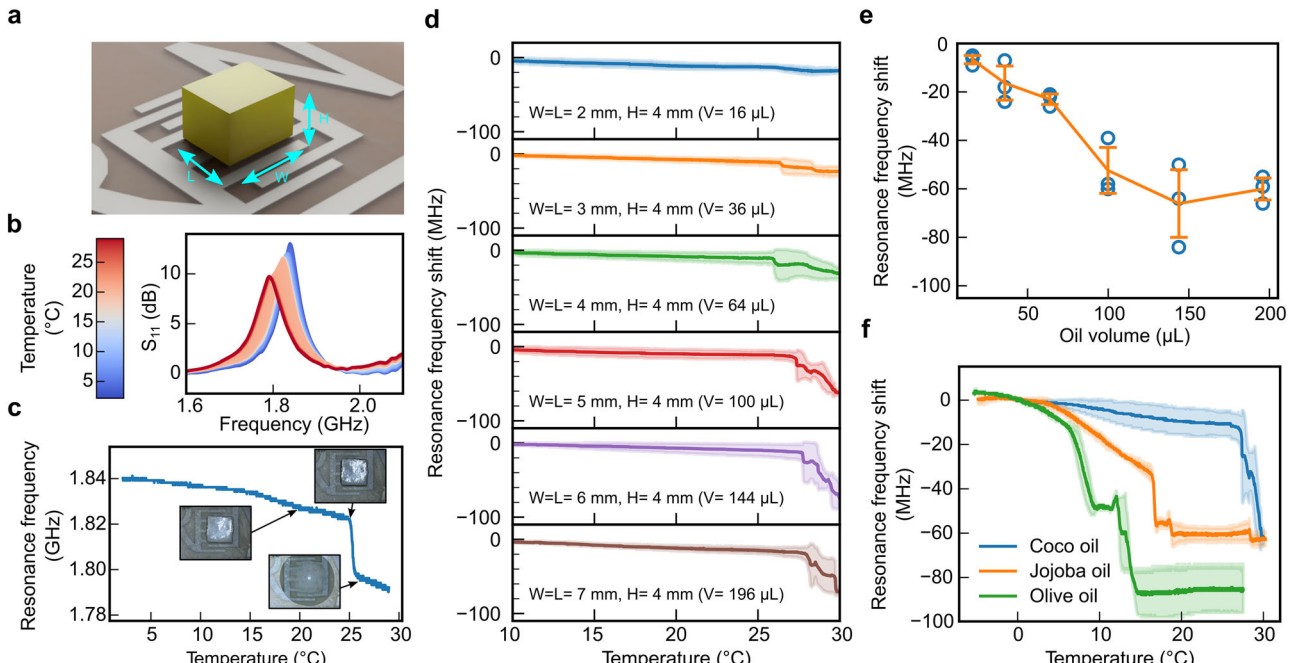

**Fig. 3 | Response after exceeding $T_{th}$ of the chipless tag when coated with the PCM. a** Width W, length L and height H of the PCM cube placed over the IDE without a paper capillary element. **b** $S_{11}$ response of the tag as temperature increases from 3 °C to 27 °C using a 4 × 4 × 4 mm³ coconut oil cube. **c** Resonance frequency change of the chipless tag as a function of temperature for the 4 × 4 × 4 mm³ coconut oil cube and optical images showing the progressive melting of the PCM. **d** Evaluating the effects of the frozen coconut oil cube size on the frequency shift after melting (n = 3). **e** Resonance frequency shift of the tag after coconut oil melting as a function of volume (n = 3). **f** Validating various threshold temperatures using 6 × 6 × 4 mm³ = 144 µL oil cubes of olive oil, jojoba oil and coconut oil (n = 3).

orientation of the package and investigate the effect of gravity. The tag and reader antenna are positioned in the climatic chamber cooled using icepacks, and the RF signal is recorded right after exposure to room temperature with a camera to track the physical state evolution over time of the PCM as temperature varies.

## Chipless tag response when coated with the phase change material

To validate the PCM melting mechanism and find the optimal volume of oil required to achieve the largest frequency shift $\Delta f$ after exceeding $T_{th}$, various amounts of frozen oil were placed over the chipless tag positioned in a flat position (θ = 0°) over the reader antenna. Allowing for simpler testing at room temperature, frozen coconut oil liquefying at 25 °C is used as PCM. Frozen cubes of coconut oil with a square base area ranging from 2 × 2 to 7 × 7 mm² for a height of 2 or 4 mm are centered over the IDE to analyse the effects of the initial coverage and volume of PCM on the $S_{11}$ response after melting (Fig. 3a). As the temperature increases, the resonance frequency of the tag shifts as the PCM melts over the IDE due to the induced changes in the dielectric ($\varepsilon_{coconut} \approx 3$) properties and RF losses, as can be seen on the $S_{11}$ response of the tag with a 4 × 4 × 4 mm³ coconut oil cube presented in Fig. 3b. This melting of 64 µL of oil leads to an irreversible resonance frequency shift of more than $\Delta f = 30 MHz$ from $f = 1.825 GHz$ at 24 °C down to $f_m = 1.785 GHz$ after exceeding 25 °C (Fig. 3c). A camera recording showing the evolution of the $S_{11}$ response of the tag as the PCM melts following an increase in temperature is available in Supplementary Movie 1.

The best coverage and volume of frozen oil over the IDE to reach an optimal response is analysed using different chipless tags samples (n = 3 for each combination of PCM base area and height tested) (Fig. 3d). A volume of oil of 144 µL, corresponding to a 6 × 6 × 4 mm³ frozen coconut oil cube centered over the IDE, provides the best response by leading to a resonance frequency shift of 65 ± 14 MHz,

while increasing the amount of oil further does not improve further the response (Fig. 3e). Decreasing the height of the PCM, to use less material and lower the device topography, leads to smaller shifts in resonance frequency (Supplementary Fig. S12) Finally, for the same optimal volume of the PCM material, by either using jojoba or olive oils to target lower threshold temperatures of 15 °C and 8 °C, respectively, the responses of the chipless tags were of 27 ± 4 MHz and 41 ± 2 MHz once exceeding the melting temperature of the respective PCMs (n = 3). These different frequency shifts observed are attributed to how the oil is melting based on its fatty acid composition and how it spreads over the beeswax, related to their surface energies and the viscosity of the PCM. A secondary unexpected frequency drop occurred for olive oil once already melted. After exceeding its threshold temperature of 8 °C, leading to the frequency drop of 41 MHz, the liquid olive oil continues to spread out over time, and at around 12 °C, seeps between the tag and the reader antenna, increasing the change in frequency to 82 ± 15 MHz, as seen on the supplementary Fig. S13. By confining the melted oil over a specific area using a circular plastic well, the induced frequency shift after transitioning of the PCM was found to be similar for all three oils, with an average frequency of resonance shift of 34.4 ± 2.8 MHz, 36.5 ± 5.3 MHz and 31.1 ± 4.4 MHz (n = 3) for olive, jojoba and coconut oil, respectively (supplementary Fig. S14). Therefore, the oil spread, impacted by the time-temperature nature of PCM transition and its interaction with the tag material, here beeswax, requires to be controlled.

## Counteracting the effects of gravity with an absorbent

The low absorbance of melted PCM by the PHBH substrate decreases measurement stability due to a gravity driven spread of the PCM resulting in a resonance shift depending on the inclination of the tag (Fig. 4a). To maintain high sensitivity after PCM melting at various tag inclination angles and prevent oil leaks over the packaged product, the tag is therefore equipped with a capillary cellulose absorbent placed

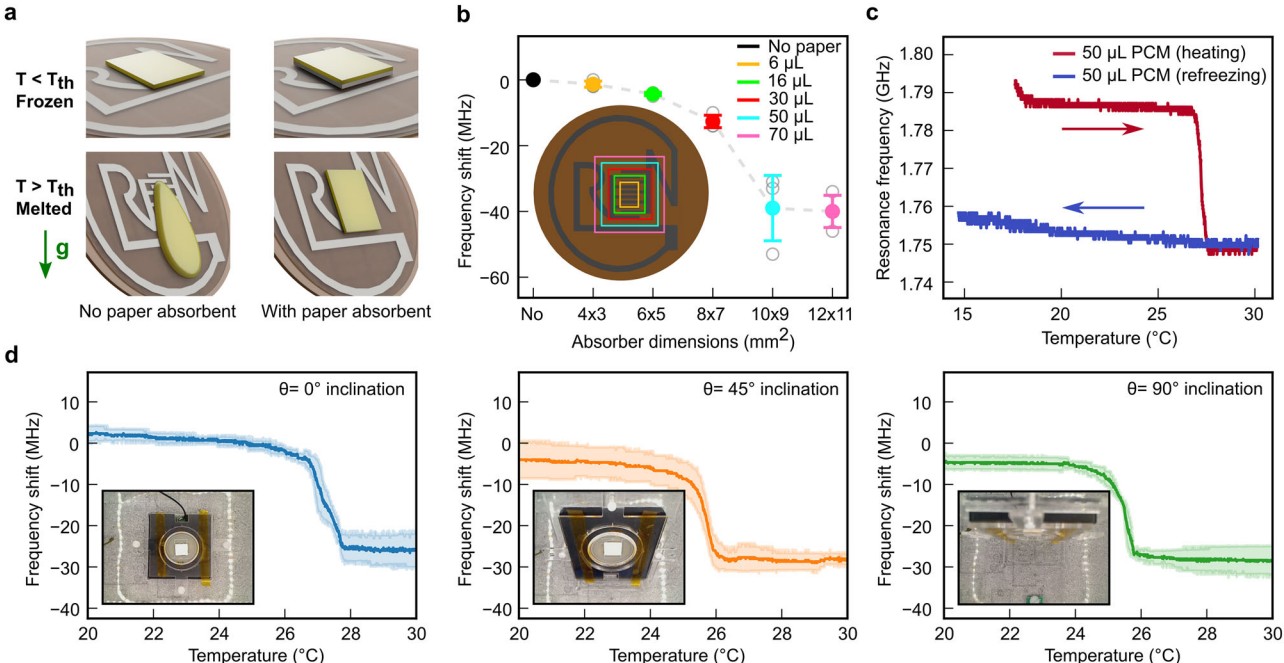

**Fig. 4 | Irreversible response of the chipless tag by counteracting the effects of gravity using a capillary paper absorbent. a** Schematic representing the effects of gravity on the melting PCM with and without paper absorbent underneath. **b** Resonance frequency shift of the tag obtained by saturating with liquid coconut oil a paper absorbent of various sizes over the IDE area ($n = 3$). **c** Resonance frequency change of the chipless tag as a function of temperature during melting and refreezing of a frozen 50 µL coconut oil element placed over a $10 \times 9$ mm² paper absorbent (Arrows indicate the evolution in temperature). **d** Resonance frequency shift of the tag with 50 µL of coconut oil placed over the paper absorbent as a function of temperature at 0°, 45° and 90° inclinations ($n = 3$).

under the frozen oil, aligned with the interdigitated electrodes of the RLC transducing layer. The cellulose capillary element was selected between 4 types of 100% cellulose Whatman filters. The one with the highest absorption capacity per area (0.54 µL/mm²) is considered to minimize its footprint (supplementary Fig. S15). The optimal dimension of this 800 µm-thick paper absorbent is selected by varying its coverage area over the IDE and considering the largest frequency shift induced between a dry and oil-saturated paper element. The largest resonance frequency shift for a coconut oil-saturated paper of $40 \pm 8MHz$ is obtained for a 9 x 10 mm² cellulose absorbent capable of retaining approximately 50 µL of melted oil ($n = 3$) (Fig. 4b). Increasing the area of the saturated paper element above 9 x 10 mm², and thus the total amount of oil possibly absorbed, had no further effect on the resonance shift as the oil is distributed further away from the IDE area. Using a more porous cellulosic capillary absorbent could enable larger amounts of oil to be used and confined over the IDE to obtain a larger shift of 65 MHz as seen previously with the 144 µL oil cube. For a substrate about 300 µm-thick, adding the paper element with this smaller volume of 50 µL of frozen oil leads to a reduction of the thickness of the PCM to 550 µm, compared to 4 mm for the 144 µL oil cube, resulting in an important reduction of the profile of the tag.

The chipless tag with the capillary paper and frozen oil slab placed over the IDE has a smaller pristine resonance frequency of $f_0 = 1.79GHz$ (Fig. 4c) due to increasing RF losses resulting from the paper element placed over the chipless tag. After exposure to rising temperatures, the resonance frequency of the tag drops by $\Delta f = 35MHz$ to $f_m = 1.75GHz$ after exceeding the coconut oil melting temperature of 25 °C. No further reduction in resonance is detected despite increasing the temperature to 30 °C, as the oil remains trapped inside the paper element thanks to capillary effects, and the shifted resonance frequency state is maintained despite refreezing the tag below the PCM melting point. The environmental parameters for testing were reproducible, as shown by the small standard deviation in the recorded temperature and relative humidity during the

experiments presented in supplementary Fig. S16. After exceeding the temperature threshold, the tag with coconut oil and paper element displays irreversible shifts in frequency of resonance very similar for tilting angles varying from 0 to 90°. The shift recorded were of $\Delta f = 30 \pm 5MHz$, $29 \pm 4MHz$ and $28 \pm 3MHz$ at 0°, 45° and 90° inclination ($n = 3$), respectively, showing that the retention of oil counteracts the negative effect of gravity (Fig. 4d). The largest standard deviation in the frequency shift after melting for the different inclinations was measured to be only 15% of the total response. The variability in the response is suspected to be due to manual fabrication of the tag, more precisely, the PHBH-cellulose substrate batch fabrication, its thickness, the printing of the zinc and the beeswax encapsulation. The variability in the response can also be due to the way the PCM melts and saturates the paper absorbent. The standard deviation is expected to decrease by integrating process automation in the manufacturing of the chipless tag. The time-temperature response of the tag containing coconut oil was also analysed under sudden exposure to 30 °C, simulating a rapid cold chain disruption (Supplementary Fig. S17). The response time of the tag depends on its initial temperature. A tag starting at −10 °C requires 140 s to show the first signs of a frequency shift, while a tag starting at 20 °C responds within just 10 s to a 30 °C exposure. Once the melting was initiated, the tag took 80 s to reach its final state. The system is therefore fast enough to detect a temperature breach before safety thresholds are exceeded[72–74]. If finer time-dependence is required, PCM and absorbent materials can be adjusted accordingly.

Ultimately, to miniaturize the reading VNA apparatus into a portable and affordable solution, a custom-made reading setup was developed to wirelessly detect the various states of the PCM-coated tag (Supplementary Fig. S18). Using principal component analysis (PCA), the portable reader can differentiate tags before and after melting of the PCM material and transfer the information to a smartphone over Wi-Fi. Furthermore, any potential impact due to the variability of the response after melting, measured to be at most 5 MHz

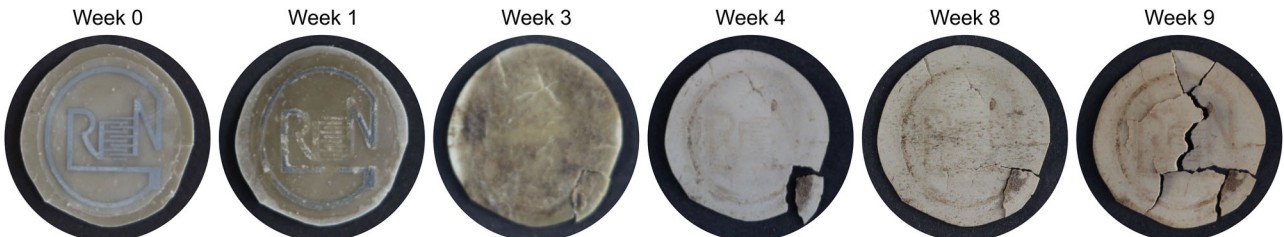

**Fig. 5 | Degradation of the eco-friendly chipless tag.** Optical images of the printed chipless tag degrading over time in a simulated compost environment according to ISO 20200, with the various pieces composing the tag recombined for the optical image.

or 15% of the total response, could be corrected by the PCA algorithm. A demonstration movie of the real-time identification of tags for three configurations (without PCM, with frozen PCM over paper and with melted PCM absorbed in the paper) is available in Supplementary Movie 2.

### Degradation of the chipless printed tag

The eco-friendly printed zinc on PHBH-50% cellulose resonating tag encapsulated with beeswax degrades in a lab-made compost environment in less than 2 months (Fig. 5). Following ISO 20200, the 15 µm-thick printed zinc making the RLC circuit is already affected by week 1. The substrate is starting to break apart in week 3, with the 100 µm-thick beeswax encapsulation disappearing one week later. After 9 weeks, the originally 375 µm-thick PHBH composite is disintegrated into multiple pieces. Reducing the thickness of the substrate would be a way to speed up the degradation of the substrate, as shown in the Fig. 1f for the 200 µm-thick composite foil.

## Discussion

In summary, we demonstrate that a chipless temperature-responsive tag for wireless cold chain monitoring can be made solely using biodegradable materials. This innovation ensures reliable performance while promoting environmental sustainability.

We showed the applicability of PHBH-cellulose as a robust, eco-friendly substrate for printed electronics and RF applications, and that the printed zinc RLC circuit on PHBH composite, encapsulated with beeswax, disintegrated in a lab-made compost environment in less than 3 months. The combination of harmless bio-based fatty acids, a cellulose absorbent to counteract gravity, and the chipless architecture of the tag enabled the wireless and irreversible detection of temperature thresholds. Other fatty acids, tuned with salts or combinations of thereof, can be envisioned to monitor different threshold temperatures of interest[62]. The current version of the tag serves as a first-line indicator for cold chain breaches. It does not provide continuous monitoring, but instead offers a simple, one-time alert that can help identify if and where in the logistics chain a temperature exceedance took place. This initial indication can prompt further investigation to assess whether the goods were affected and to determine the cause of the breach. A new chipless tag can then be applied for further monitoring during the remainder of the transport. For applications requiring precise time–temperature tracking or more complex response behavior, the system could be redesigned with tailored PCMs with specific thermal profiles. The protection of the tag and its attachment to packaging remains to be tackled. The tag could be integrated with eco-friendly adhesives and a cover to prevent the deterioration of the temperature-sensitive PCM, but still allowing its spread when melting. Table 1 below provides a comprehensive comparison of chipless resonators designed for detecting the exceedance of specific temperature thresholds. The comparison includes key parameters such as fabrication method, substrate and conductor materials, sensing mechanism, temperature thresholds, frequency shifts, and eco-friendliness. Notably, this work is the only fully

degradable in compost, chipless thermal indicator reported to date for irreversible detection of temperature thresholds. Unlike other systems, which are either partially eco-friendly or do not report long-term stability and variability of the response, the proposed chipless tag combines sustainability, reproducibility, and operational reliability. The use of biodegradable substrate made of PHBH-cellulose, degradable printed zinc and natural oils, along with the stability of the frequency shift and reproducibility of results, positions this work as a significant advancement over the current state-of-the-art.

This system is designed for applications in the perishable goods supply chain as a simple one-time indicator of temperature-exceedance in the cold chain. Our sustainable approach provides low-cost, chip- and battery-free deployment at scale in comparison to conventional solutions that offer continuous monitoring and data logging but require embedded power supply and more consequent integration. Moreover, looking specifically at active and passive RFID devices, our chipless design removes the need for silicon-based circuitry, reducing both cost and environmental impact, and making it better suited for disposable and high-volume applications. Compared to colorimetric indicators, it offers desired performance for a similar cost and easy integration in packaging while avoiding optical sighting issues like subjective interpretation.

In terms of system implementation, having an array of such chipless tag elements with bio-based PCMs tuned for different melting temperatures could enable multi-threshold detection in an all-in-one solution. As presented in the manuscript, the interrogation range of less than 20 mm currently limits the realistic simultaneous detection of multiple tags. For that matter, the antenna footprint or emitted power would have to be increased. According to the application scenario, it would be of interest to implement identification and extend the read-out range for a simpler tracking of the products, which would demand to modify the chipless tag design. Notably, a chipless L-slot design, which passively encodes binary identification, could be incorporated in the array to provide a unique long-range identification of the products in the supply chain[75]. Finally, while most reader antennas for chipless technology still need to be connected to a bulky and expensive VNA[76], our custom-developed reader offers a cost-effective alternative to traditional laboratory equipment, with a total cost comparable to a mid-range smartphone (under $400). The integrated nanoVNA, which facilitates S-parameter readout, is priced at approximately $200, while the Beaglebone used to run the Python PCA script costs less than $100, at the time of writing. Importantly, the reader has been designed to enhance accessibility and practicality, incorporating embedded Wi-Fi functionality that allows seamless interaction with smartphones, which are widely used in logistics and smart packaging applications. This functionality, demonstrated in the accompanying Supplementary Movie 2, highlights the potential for real-world implementation. Nevertheless, further development of alternative, cost-effective readout systems for long-range communication, leveraging advanced signal processing to mitigate environmental interference, remains an important future direction.

**Table 1 | Summary of chipless resonators used for temperature-threshold excursion detection**

| Reference | 78 | 60 | 61 | 64 | 63 | This work |
|---|---|---|---|---|---|---|
| Fabrication | Etched | Etched | Etched | Etched | Printed | Printed |
| Substrate (area, thickness) | Taconic TLX_0 ($6 \times 6$ mm², 0.5 mm) | Taconic TLX_9 ($20 \times 20$ mm², 2 mm) | Taconic TLX_8 ($25 \times 25$ mm², 0.5 mm) | Taconic TLX_8 ($36 \times 36$ mm, 0.5 mm) | Paper glued on blotting paper ($30 \times 30$ mm²) | PHBH-Cellulose (20 mm diam., 0.3 mm) |
| Conductor | Copper | Copper | Copper | Copper | Silver | Zinc |
| Sensing material (mechanism) | Phenanthrene (Sublimates) | NaCl-water (Uncontrolled melting) | Grapeseed & coconut oils (Uncontrolled melting) | Grapeseed, coconut & palm oils (Uncontrolled melting) | Grapeseed & coconut oils (Controlled melting) | Olive, jojoba & coconut oils (Controlled melting) |
| Degradable materials | No | Partially | Partially | Partially | Partially | Fully |
| Temperature threshold (°C) | 72 | −15 to −1 | 4, 8, 12 | 14.2, 17.3, 23.6 | 19.7, 21.9 | 8, 15, 25 |
| Resonator | RLC | Ladder-shape (time-based) | Circular Asymmetric SRR | Circular Asymmetric SRR | Square SRR | RLC |
| Operating frequency (GHz) | 6 to 7 | 3.5 to 5.5 | 3 to 6 | 3 to 5 | 2 to 4 | 1 to 5 |
| Frequency shift (MHz) | 320 | NA | 250 | 400–600 | 250, 600 | 30–40 |
| Variability of frequency shift | NA | NA | NA | NA | NA | 5 MHz or 16.6% |
| Stability | NA | NA | NA | NA | NA | After 1 year, 4 MHz drift and >30 MHz response |
| Response time | 60 min | NA | NA | NA | 30 min | Few to 140 s* |
| Disintegration in compost | NA | NA | NA | NA | NA | Yes |

*Depending on the temperature difference experienced by the tag.

## Methods

### Commercial electronic paper

The commercially available printed electronic paper (PE paper) used was the PowerCoat HD with a thickness of 230 µm (Fedrigoni, France).

### Fabrication of the nanopaper substrate

Nanofibrillated cellulose (CNF) was produced according to a well-established protocol from Saito et al. and from elemental chlorine-free cellulose fibers from bleached softwood pulp (Stendal GmbH, Germany) (functionalized with NaClO at 10 mmol per gram of cellulose pulp)[77]. The solution was fibrillated using 10 kWh (Mazuko Super-collider, Japan). The mass ratio of fibers and CNFs were calculated to achieve a nanopaper with an areal density of 120 g m$^{-2}$ (80 g/m$^2$ for 100% CNF). The CNF suspension was mixed for 30 min at 1200 rpm (IK T45, Ultra Turrax USA), followed by filtering and sheet forming twice at 95 °C.

### Fabrication of the PHBH-cellulose composite substrate

Cellulose fibers for the composite were processed from a 30 µm-sized cellulose powder (BE-600/30 Arbocel, Germany). The poly(3-hydroxybutirate-co-3-hydroxyhexanoate) (PHBH) containing 11 mol % of hydroxyhexanoate was obtained from the MAIP Group in pellet form. 0.2 g of Dicumyl peroxide (DCP) (Sigma Aldrich) is dissolved in acetone in a three-neck flask with a stirrer for 10 min, 10 g of cellulose is gradually dispersed in the acetone-DCP solution and left 30 min to stir. The acetone was removed from the suspension in a vacuum oven for 12 h. 0.1 g of DCP is mixed with 10 g of PHBH and then mixed with the cellulose coated with the DCP in powder form. The powder composite was compounded twice in a twin-screw extruder at 175 °C (MiniLab 3 Micro Compounder from Thermo-fischer, Germany).

### Profilometry and mechanical testing of PHBH-cellulose composite

The substrates were stored at 22 °C and 50% relative humidity over-night. Square map scans of different lengths (50 µm, 100 µm and 300 µm) were performed (Bruker Dektak XT, Switzerland), with a range set to 526 µm, a tip radius of 2 µm and a force of 3 mg and a resolution of 0.5 µm/pt.

The samples were conditioned at 20 °C and 50% relative humidity overnight. They were prepared by punching the substrate into dog-bone shapes and tested under tension in a universal pull testing machine (AllroundLine by Zwick, Germany). The dog-bones were pulled at a loading velocity of 0.8 mm/s until rupture.

### Dynamical vapor sorption of substrates

The samples were conditioned at 20 °C and 35% relative humidity overnight. The tests were performed at 40 °C from 30% RH to 70% RH, with 10% RH steps with conditions to go to the next step: duration exceeding 3 h per step or a mass variation less than 0.04% for 5 min for the nanopaper, and PE paper; duration exceeding 8 h per step, or a mass variation less than 0.04% for 20 min for pure PHBH and PHBH-cellulose composite.

### Thermal and moisture stability of pure PHBH and 50% cellulose composite substrates

To investigate the effect of moisture on the mechanical stability of the substrates, silver ink-based (Suntronic, AST6025, Sigma Aldrich) resistors were screen printed using a 90–40w PET mesh. For all the substrates, the printed tracks were then cured at a low temperature of 40 °C overnight to avoid any thermal influence on their mechanical properties. The samples were first left 4 h at 30% RH, followed by climatic cycling at 30-40-30-50-30-60-30-70% RH during 2 h for each step with a 30 min ramp in between. The test was performed at 25 °C, and the electrical resistance of the resistors printed on the various substrates was probed using two-point measurements every 3 s and averaged over a period of 3 min at the end of the RH step.

Differential scanning calorimeter analysis was conducted for the pure PHBH and the PHBH-50% cellulose composite. 15 mg samples were prepared and heated from −50 to 200 °C. Following this initial heating to reset the thermal history of the material, the substrates were cooled down to −50 °C and then reheated to 200 °C following a 20 °C/min ramp. Heat flow data was taken every 0.5 °C. Thermogravimetric analysis was conducted from 0 °C to 900 °C for both pure and 50% cellulose composite PHBH samples. Weight and temperature data were acquired every 30 s.

For the thermal sintering compatibility of the PHBH composite, commercially available silver ink (5065 DuPont) was screen printed over PHBH-50% cellulose. The printed resistors were sintered at either 120, 130, 140, 150 °C for 30 min in an oven and the resistance of the metallic trace was measured using 4-point probing ($n = 3$). The changes in appearance of the printed test sample and its mechanical behavior after oven curing were analyzed optically to assess the thermal limits of the substrate.

### Printed microstrip line RF characterization

Commercially available silver ink (5065 DuPont) was printed over PE paper and PHBH-50% cellulose. The microstrip lines were sintered at 130 °C for 30 min. The microstrip line was connected to a Vector Network Analyzer (Agilent E5071C), and the device under test was placed in a climatic chamber for 3 h at 30% RH and 3 h at 70% RH. The radio frequency $S_{12}$ signal of the printed microstrip line was recorded from 1 GHz to 3 GHz every 5 s for 5 min at the end of both relative humidity steps and averaged.

### Substrate disintegration

We followed the international standard ISO 20200 to evaluate the disintegration rate of the 200 µm-thick pure PHBH and 50% cellulose composited substrates under simulated aerobic composting conditions in a laboratory-scale test. It was carried out in soil at 58 °C for 28 days. The soil was composed of sawdust, rabbit deed, cornstarch, sugar, corn oil, urea, sieves and compost. We put the sample inside a protective mesh and buried it completely in the soil. Optical pictures of the sample were taken in weeks 1, 2, 3 and 4.

### Printed eco-friendly chipless tag fabrication

The zinc ink was made using spherical nanoparticles (300 nm, 99.9% purity, US Research Nanomaterials), polyvinylpyrrolidone (PVP) (Mw = 360 K, Sigma Aldrich) and pentanol (99%, Sigma Aldrich) in a 25:1:5 weight ratio and mixed for 30 min at 300 rpm (Thinky ARE-250). Printed over a 300 µm-thick PHBH-cellulose composite substrate by screen printing, the zinc was chemically sintered by spraying 10% acetic acid in deionized water, followed by three photonic sintering pulses at 225 V at 0.1 Hz (Novacentrix 1200, PulseForge). The now conductive zinc was encapsulated with a 100 µm-thick layer of bees-wax (Sigma Aldrich) by melting a disk at 80 °C during 30 s. The 20 mm-wide beeswax disk was created by melting 0.04 g of wax into a mold at 80 °C, left to cool down on a flat surface at room temperature. After 1 h, the mold was placed at −18 °C in the fridge overnight for release.

### Phase-change materials and absorbent

Olive oil, jojoba oil and coconut oil (Sigma Aldrich) were heated to 60 °C for 30 min to ensure proper melting and poured into silicone molds of various volumes to form cubes of base area ranging from 2 x 2 to 7 x 7 mm$^2$ for a height of either 2 or 4 mm. The PCM cubes were frozen at −8 °C before use. The desired frozen cube was placed over the IDE, and the resonance frequency of the tag was measured as the oil melts when exceeding its threshold temperature.

100% cellulose absorbents for oil retention (Whatman 1001, Whatman 3, Whatman 3MMCHR and Whatman GB003) were laser cut

in circles of different radii from 2.5 mm to 10 mm (Speedy 3000, Trottec) and fully saturated with liquid jojoba oil at room temperature with excess oil being removed with a cloth. The soaked samples were weighed to identify the amount of oil absorbed ($\rho_{jojoba}$ = 0.87 g/mL, at RT) per absorbent area at saturation to select the best paper absorbent.

To assess the optimal paper dimensions over the chipless tag, 800 μm-thick Whatman GB003 rectangles are laser cut in sizes ranging from 4 x 3 mm² up to 12 x 11 mm². The pristine (no oil) or saturated with coconut oil paper rectangles are centered over the IDEs. The soaked paper elements were filled with oil volumes corresponding to their size, from 6 to 70 μL, considering the absorption capability of 0.54 uL/mm². The resonance frequencies of the tag with the paper elements are recorded at RT, and the frequency shifts obtained between the dry and soaked paper absorbents are computed.

For temperature-based measurement, the frozen PCM and absorbent were stacked, and an eco-friendly glue made of 20% Poly-Vinyl Alcohol (Sigma Aldrich) in deionized water was used to fix them over the beeswax-coated wireless tag.

### Wireless testing of the tag in simulated cold chain condition
The fabricated tag was placed over a custom-made copper on FR4 near-field reader antenna at 2 mm distance using a spacer. The tag and reader antenna were placed inside a non-metallic custom-made climatic chamber to prevent interference. The reader antenna was connected to a vector network analyzer (Agilent E5071C). The $S_{11}$ magnitude of the reader antenna was measured as a reference and subtracted from the recorded $S_{11}$ signal with the tag above the antenna. The $S_{11}$ signal of the tag was sampled every 5 s in the frequency range of 1.5 – 3 GHz with a resolution of 1 MHz. The temperature and humidity of the chamber were monitored every second with a commercially available environmental sensor (SHT30 from Sensirion AG). The chamber was cooled down below 0 °C using icepacks and suddenly exposed to room temperature to simulate a breach in the cold chain. To reach temperatures above RT, a hot air gun was implemented to slowly heat up the testing chamber. The temperature ramp was approximately 0.2 °C/min.

### Wireless identification of the state of the tag
The custom-made portable reader antenna is composed of three elements: a nanoVNA, operating up to 4 GHz, a custom loop antenna and a BeagleBoard controlling the nanoVNA. Principal component analysis was applied for the detection and identification of different tags positioned over the reader antenna. Python implementations were developed to handle both consecutive numerical identifiers, where histograms are stored as Python lists, and arbitrary identifiers, where histograms are stored as dictionaries. The Python script is executed on the BeagleBone embedded within the custom-designed reader. The PCA method effectively separates and identifies different tag states, as illustrated in the accompanying Supplementary Movie 2. The algorithm was trained by repeatedly position tags 5 times over the reader antenna and analyzing the resulting $S_{11}$ response based on the presence and state of the coconut oil material. The training and demonstration were performed at room temperature for the three different configurations considered, namely the pristine tag without paper element and PCM, the tag with paper and frozen PCM, and finally the tag with paper soaked in oil following its melting.

### Printed chipless tag disintegration
We followed the international standard ISO 20200 for determining the degree of disintegration of plastic materials under composting conditions in a laboratory-scale test to evaluate the disintegration rate of the zinc printed tag on PHBH-50% cellulose composite with beeswax passivation under simulated aerobic composting conditions. It was carried out in a laboratory-made soil, at 58 °C for 10 weeks. The soil was composed of sawdust, rabbit deed, cornstarch, sugar, corn oil, urea, sieves and compost. We put the sample inside a protective mesh and buried it completely in the soil. Optical pictures of the sample were taken every week, and once shattered into pieces, the disintegrated tag was recombined for the picture.

### Chipless tag and reader antenna simulation
Commercially available full-wave electromagnetic solver, Ansys HFSS, was employed for the design, optimization, and theoretical performance estimation of both the reader antenna and the multi-resonant tag structure. The simulation environment enabled accurate modeling of the antenna geometry, substrate characteristics, and near-field coupling behavior critical to short-range operation. Parametric sweeps and optimization routines were used to tune the antenna dimensions for wideband impedance matching and stable radiation performance across the multiple resonances of the tag. Theoretical estimations, including input reflection coefficient ($S_{11}$), near-field distribution, and antenna gain, were validated within the HFSS environment under realistic boundary and excitation conditions.

## Data availability
The data that supports the findings of this study is available on the repository Zenodo under the following access code: 10.5281/zenodo.16920654.

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

## Acknowledgments

The authors kindly acknowledge funding from the Swiss National Science Foundation and Innosuisse BRIDGE Discovery program for the project "GREENsPACK—Green Smart Packaging" (Grant No.: 187223).

## Author contributions

D.B., G.N., and O.V. obtained the funding and managed research at EPFL, EMPA and CSEM. X.A. and G.S. manufactured the PHBH substrates. X.A., G.S., and J.B. studied the substrate properties. J.B., N.F., and J.K. manufactured the chipless tags. J.B. and N.F. prepared the PCM materials and characterized the tags. X.A. and G.S. conducted the disintegration tests. J.B., N.F., J.K., X.A., G.S., O.V., C.B., and D.B. analyzed the results. O.V. and C.B. conducted the simulations. All authors contributed to and reviewed the manuscript writing. D.S. helped coordinate the project. All authors have given approval to the final version of the manuscript.

## Competing interests

The authors declare no competing interests.
