## [Transparent Peer Review file · Nature Communications]

Ecoresorbable chipless temperature-responsive tag made from biodegradable materials for sustainable IoT

Corresponding Author: Dr James Bourely

Version 0:

Reviewer comments:

Reviewer #1

(Remarks to the Author)

In this manuscript, Bourley et al. proposed an ecoresorbable chipless temperature sensing technology for cold chain temperature monitoring. The author has indeed presented a relatively novel concept and undertaken a distinctive approach; however, the consideration of details has been insufficient, whether the present design is suitable as a sensor is a question. Below are my detailed comments:

Just as a thermistor will respond to changes in temperature, a change in temperature will inevitably result in a corresponding change in the thermistor's resistance. However, this reasoning does not establish that the sensor has been successfully developed. Likewise, differentiating various states due to changes in the sensor materials' properties is straightforward. However, can the reverse be achieved? That is to say, can the frequency spectrum readings be used to infer the sensor's state? What is the confidence level associated with this inference?

From a quantitative perspective, what is the reproducibility of the experiments? Is there a direct correlation between frequency and the state of the tag? Will manufacturing precision and variability between different batches of materials influence the final frequency measurements?

The manuscript should provide a comparison of the proposed technology's performance against currently available commercial products, focusing on key metrics such as sensitivity, accuracy, resolution, long-term stability, and response time.

"What is the definition of 'irreversible temperature changes'? In the respective context, irreversible changes refer to food spoilage rather than temperature alteration. Temperature is a parameter that is always reversible concerning changes in environmental conditions.

Is 'zinc metallic traces' considered part of "exclusively biodegradable materials"?

How significantly does the rate of temperature change impact this sensor? If there is a momentary spike in temperature, what type of response would the sensor exhibit?

I understand that the author intends to utilize phase change materials (PCMs) for material modification due to elevated temperatures over specific durations, reflecting this through changes in the material's resonance frequency. However, traditional temperature sensors are more suited for long-term continuous temperature recording. In the current design, once the phase change of material happened, it no longer can restore to its original state.

The author claim that the sensor can be used for sustainable IoT but only demonstrated its potential application in cold chain temperature monitoring. First of all, NFC communication has very limited communication distance, to what extent can this technology be useful for IoT? How close must the PCM be to the NFC reader to obtain an accurate measurement? Secondly, The author has only demonstrated initial results regarding the materials' performance under environmental temperature changes. However, the practical utility of the tag in real-world applications has not been adequately demonstrated." Furthermore, what is the dimension of the sensor? Is the present design suitable to be integrated to cold chain temperature monitoring? Can the sensor be significantly miniaturized?

The author has stressed that the sensor is "chipless". Please further elaborate this point by: (1) Explain what chips are used from conventional temperature sensor? And why the chipless design is preferred over the conventional design? (2) This present sensor design still applies NFC reader which contains chip to extract data, what is this sensor considered chipless?

Reviewer #2

(Remarks to the Author)

The authors developed a fully biodegradable, chipless, and wireless temperature sensor tag that uses eco-friendly phase-

changing materials to detect temperature thresholds in cold chain monitoring.

The paper is well-written and well-presented, including the accompanying figures. While the concept is ingenious, it is not entirely novel; the approach of modifying the resonant frequency of an RLC circuit based on changes in substrate permittivity has been widely explored in previous studies. However, what I found most commendable is the development and fabrication of a biodegradable substrate and ink for the tag.

That said, I am not fully convinced that the work is suitable for publication in such a high-impact journal, although it does have merit. If the following comments and suggestions are addressed, the work could be enhanced and potentially meet the standards of this high-quality journal:

- First of all, I believe two important aspects are missing and should be addressed: 1) A comparison of your developed system with other similar systems previously published, and based on that, 2) An explanation of what characteristics, or combination of characteristics, make your system unique and advance beyond the current state of the art.

- Although the use of biodegradable materials is a clear advantage from a sustainability perspective, it also limits the operational lifespan of the tag. In fact, the printed zinc affects the performance of the RLC circuit as early as week 1. Therefore, how long can the tag remain functional before becoming ineffective due to biodegradation? This limitation significantly restricts the potential applications of the system in real-world scenarios and should be clearly addressed in the manuscript.

- One aspect that has particularly caught my attention is the incorrect conceptual use of Near Field Communication (NFC) technology in this work. The authors frequently refer to NFC communication, including terms such as "chipless NFC tag," "NFC reader," and "NFC antenna." However, this is inaccurate. Firstly, NFC technology operates at a frequency of 13.56 MHz, whereas the developed tag functions in the GHz range. Secondly, NFC is a set of communication protocols, which makes the term "chipless NFC tag" fundamentally incorrect, as these protocols require a chip to facilitate communication. For example, the communication between an NFC tag and an NFC-enabled smartphone is made possible through these protocols. It is therefore misleading to state that the authors have developed a custom NFC antenna or reader. Additionally, it is quite surprising that the custom reader purportedly detects and reads the chipless tag, yet then uses an entirely different wireless protocol (WiFi) to transmit the data to the smartphone, as demonstrated in the supplementary video. Clearly, NFC is not the technology being employed in this work. I kindly request that this significant conceptual error be addressed and corrected.

- I was also surprised by the use of Machine Learning by the reader to differentiate between the tags. Is it truly necessary to employ ML for identifying only three different resonance frequencies? It seems excessive for such a purpose.

- Although the concept is ingenious, the need for results to be obtained using bulky and expensive laboratory equipment (e.g., a VNA) or a custom-developed reader significantly limits its potential application in real-world scenarios, such as smart packaging. While the authors acknowledge this limitation in the Discussion section, I believe it poses a substantial constraint that could reduce the likelihood of publication in very high-impact journals.

In summary, the work is very interesting, particularly the development of the entire tag using biodegradable materials. However, the novelty of the work is not entirely clear, and there are several limitations that affect the practical usability of the developed system. Additionally, some significant conceptual errors are present. These factors make me hesitant to determine whether this work is suitable for publication in a high-impact journal like Nature Communications. Nevertheless, addressing the aforementioned points would be necessary to consider the possibility of publishing in such a high-quality journal.

Reviewer #3

(Remarks to the Author)

The manuscript presents an innovative approach to developing a chipless temperature sensing tag using biodegradable materials, which is a commendable effort towards sustainable IoT solutions. The concept of using eco-friendly materials is highly relevant and timely, given the increasing focus on environmental sustainability.

However, there are several critical issues that need to be addressed to enhance the manuscript's clarity and scientific rigor. Firstly, the temperature sensitivity of the chip, which starts shifting frequency at 8, 15, and 25°C, raises concerns about handling and transportation before use. It is unclear whether these chips need to be stored in a freezer to avoid transition. Additionally, the sensor's decomposition within 14 days limits its practical application in the supply chain due to its short lifespan.

Consistency in terminology is another issue, with terms like CFN paper and nano paper, as well as NFC antenna, reader, reader antenna, and antenna, being used interchangeably. This inconsistency can confuse readers and detract from the manuscript's overall coherence.

Despite these concerns, the manuscript provides valuable insights into the potential of biodegradable materials for IoT applications. The innovative use of PHBH/Cellulose paper and the detailed analysis of its properties are noteworthy. However, the manuscript would benefit from addressing the following points:

- 1.State of the Art in Chipless RFID: Include a review of similar research works and publications to highlight the novelty and

scientific contribution of the proposed work.

2. Figure 1 Clarification: Ensure that the papers in Figure 1 are of the same thickness as the 200-micron PHBH/Cellulose paper, as variations in thickness can significantly impact absorption and S11 response.
3. Water Vapor Sorption Test: Clarify why 30-70% RH was selected and how it correlates to RH in a cold chain environment.
4. The manuscript should also specify the temperature used in the test and its relevance to actual cold supply chain conditions.
5. RF Measurement Conditions: Provide details on the temperature during RF measurements and the relationship between temperature and RH. Experimenting with the impact of temperature while keeping RH constant is strongly suggested.
6. Decomposition Rate: Calculate the decomposition rate considering the initial mass of the paper, as larger or thicker paper will take longer to decompose.
7. Chipless Tag Design and Test Setup: Justify the change in thickness from 200 to 304 microns and provide experimental support for the claim that 304 microns minimises undesirable bending. Further tests are needed to determine the thickness threshold for minimal bending.
8. Impact of Substrate Deformation: Address how the proposed RFID sensor minimises the impact of mechanical deformation on frequency shifts.
9. NFC Antenna Usage: Clarify the discrepancy between using an NFC antenna for RF measurement, given the different operating frequencies.
10. RF Measurement Setup: Improve the RF measurement setup to follow standard practices, such as using an anechoic chamber to minimise interference and provide reliable results.
11. Oil Frequency Shift Analysis: Explain why olive oil induces the largest frequency shift compared to other oils and provide a critical analysis of this observation.
12. Beeswax Impact: Test and report whether beeswax affects RF performance and its mechanical properties. Absorbance Test: Include the relationship among the absorbance of the filter paper, its mass, the volume of oil, and the contact area of oil.
13. Machine Learning Justification: Justify the use of machine learning for a simple differentiation task and provide technical details about the approach. If this section does not contribute to the paper, consider removing it.
14. Supplementary Figure 7a: Clarify why both pure PHBH and 50% PHBH/cellulose materials have a trough around 50°C in the DSC graph.

Overall, while the manuscript does not provide significant advancements in the field, it does offer a novel approach to sustainable IoT solutions. Addressing the aforementioned issues will enhance the manuscript's scientific rigor and clarity, making it a more valuable contribution to the field.

Version 1:

Reviewer comments:

Reviewer #1

(Remarks to the Author)

Thank you for addressing my previous comments and revising the manuscript. The authors have made commendable efforts to improve the clarity and scope of the work. However, several critical issues remain unresolved and require further attention to ensure the manuscript meets the high standards of Nature Communications. Below are my detailed concerns and recommendations:

1. Terminology: "Temperature Sensor" is Misleading

The manuscript continues to describe the device as a "temperature sensor." However, the system does not fulfill the technical definition of a sensor, which is expected to reversibly quantify thermal energy and convert it into a measurable signal. Instead, the presented device operates irreversibly via phase-change material (PCM) properties. To avoid misinterpretation, I strongly recommend replacing "sensor" with a more accurate term such as "temperature-responsive tag," "thermal indicator," or "PCM-based monitor" throughout the manuscript. This revision is critical to maintain scientific rigor and prevent confusion among readers.

2. Statistical Significance of Frequency Response Variability

The authors note a 15% standard deviation in the frequency response across repeated tests (as stated in the response letter). While this provides some insight into reproducibility, it is suggested that the manuscript:

- Explicitly include this value in the Results/Discussion section.
- Justify whether 15% variability is acceptable for the intended application (e.g., by benchmarking against industry standards or comparable studies).
- Discuss potential sources of variability (e.g., fabrication tolerances, environmental factors) and strategies to mitigate them.

3. Citations for Zinc as a Bioresorbable Material

The use of zinc as a degradable material is central to the device's design. However, the manuscript lacks references supporting zinc's bioresorbability and degradation mechanisms. Relevant citations should be added to strengthen this claim and contextualize the material choice.

4. Practical Limitations of Response Time and Economic Viability

The authors' new experiments reveal that the system's response speed depends on the PCM's initial storage temperature. While this is an important finding, the manuscript is suggested to address:

- **Applicability Concerns:** How does the response time align with real-world scenarios (e.g., food spoilage timelines)? If the tag's response is slower than the spoilage process, it may fail as a safety indicator. Conversely, if it responds too quickly, it risks premature inactivation.
- **Economic/Technical Comparison:** How does this system compare economically and functionally to existing temperature sensors (e.g., cost, ease of deployment, reusability)? A brief discussion of trade-offs would clarify the technology's niche.

5. Feasibility of Near-Field Operation (2 mm Range)

The manuscript states that the tag requires a 2 mm distance from the reader for operation. For real-world adoption, this limitation raises questions:

- How would such a short range be practically implemented (e.g., integration into packaging, handheld readers)? If a handheld reader is to be used, how would a large number of tags be monitored simultaneously?
- Are there plans to improve the readout range, or is this proximity intentional (e.g., to prevent interference)?
- A figure or schematic illustrating the envisioned use case (e.g., placement on food containers) would help readers visualize the application.

Conclusion

This work presents a novel approach to temperature monitoring, but the above issues must be resolved to ensure the manuscript is technically sound and accurate. I recommend major revisions to address these points before further consideration.

Please let me know if further clarification is needed.

Reviewer #2

(Remarks to the Author)

The authors have successfully addressed all my comments and suggestions. Congratulations for the conducted work.

Reviewer #3

(Remarks to the Author)

I have reviewed the responses to all comments and revised the paper accordingly. A few additional comments are made to ensure the clarity of the paper.

R2C7 (Response to comment 7): Bending Test The authors have conducted further experiments to demonstrate the impact of bending on the performance of the antenna. However, the experiments require more depth. For example, the results presented in Figures S5 and S6 are based on only one bending cycle, resulting in a very small frequency shift. In a more realistic environment, the tag might be subjected to multiple bending cycles, potentially amplifying the frequency shift. Therefore, it is necessary to observe the frequency shift after multiple bending cycles at each radius. Consistency could be demonstrated through this experiment.

R2C9: Real Environment Testing It is understood that the tag is intended for use in a real environment, making real environment testing adequate without using an anechoic chamber. While it is true that the antenna has been benchmarked, this does not apply to the proposed antenna, which has yet to be benchmarked despite its small reading range of 2mm. Therefore, a thorough investigation of the antenna, including 2D and 3D radiation patterns, is required to determine the transmission pattern of the antenna.

R2C11: S11 Reduction As shown in Figure S1, the S11 of the antenna has significantly reduced from -25dB to -8dB after one year of storage with a beeswax encapsulation. What could be the cause of this reduction, and is -8dB sufficient for the reader to pick up the tag repetitively?

Addition comment: New Table 1 Reference 56 seems to produce a large frequency shift of 600 MHz compared to the present work's 40 MHz. Is this because their antenna is more sensitive? Additionally, the term "Reproducible frequency shift" is introduced, and the present work claims to have a reproducible frequency change over other work. However, there is no data to support this claim. In addition, it would be more useful to include a comparison of S11 at their resonant frequencies

and the frequency shift due to bending for all included work.

Typo in Table 1 Should the resonator be RLC, not ELC?

Version 2:

Reviewer comments:

Reviewer #1

(Remarks to the Author)

Thank you for the thorough response. I appreciate the additional revisions, which have clearly improved the quality of the submission. While I acknowledge that the current implementation still faces practical limitations—particularly regarding readout distance, device robustness, and real-world applicability—I recognize the novelty of the approach and its potential value as a platform for further development. Therefore, after careful consideration, I am pleased to accept the manuscript for publication in Nature Communications.

Reviewer #3

(Remarks to the Author)

I am pleased to see that my previous comments have been thoroughly addressed through the addition of new experiments and simulations. These enhancements significantly strengthen the manuscript by providing more comprehensive evidence to support the central hypothesis.

I have only one minor suggestion: in Figure S7, the current color scheme, particularly the use of blue, makes it somewhat difficult to distinguish between the curves. To improve clarity, I recommend adding distinct symbols or markers to each curve, which would enhance the figure's readability and overall presentation.

Point by point response to reviewers:

Reviewer 1

Comments:

1. Just as a thermistor will respond to changes in temperature, a change in temperature will inevitably result in a corresponding change in the thermistor's resistance. However, this reasoning does not establish that the sensor has been successfully developed. Likewise, differentiating various states due to changes in the sensor materials' properties is straightforward. However, can the reverse be achieved? That is to say, can the frequency spectrum readings be used to infer the sensor's state? What is the confidence level associated with this inference?

We believe that the use of NFC may have led to a misunderstanding of the working principle of the tag. The proposed chipless tag operates based on two distinct resonance frequencies, depending on the two different potential states of the PCM material on the tag. When the tag with the PCM is below the temperature threshold, the PCM remains frozen on the interdigital electrodes (IDE), causing minimal effect on the resonance frequency. However, once the temperature threshold is exceeded, the PCM melts and is absorbed into the paper absorbent, introducing additional dielectric losses and causing a significant reduction in the resonance frequency, i.e. to answer the question it is indeed possible to use the frequency spectrum reading to deduce the state of the sensor. Due to capillary effects and the oil being trapped in the absorbent, the reverse behaviour is not possible, as seen in Fig. 4c. In this graph, the frequency of resonance after melting of the PCM and returning to 15 °C (or 10 °C below the melting point of the PCM) is still more than 30 MHz. On the other hand, a further temperature increase above the threshold temperature of the sensor will continue to lower the resonance frequency, as the resistance of the zinc track making the LCR circuit will rise due to the positive temperature coefficient of resistance (TCR) of zinc ($\alpha = 0.0038$).

We are confident in the ability to detect the crossing of a temperature threshold, as demonstrated by the reproducible inclination experiments shown in Fig. 4d, where a maximum standard deviation of 15% is measured on the frequency response across multiple tests. Furthermore, the 30 MHz frequency shift, maintained even after cooling the tag back by 10 °C below the threshold temperature, boosts our confidence. A newly conducted experiment detailed when addressing the next comment resulted, again, in a consistent frequency shift of more than 30 MHz after PCM melting obtained for tags with olive, jojoba and coconut oils (Fig. S10 detailed in the next comment).

2. From a quantitative perspective, what is the reproducibility of the experiments? Is there a direct correlation between frequency and the state of the tag? Will manufacturing precision and variability between different batches of materials influence the final frequency measurements?

Firstly, the environmental parameters recorded during the experiments were not presented in the first version of manuscript. Therefore, we have added the recording of temperature and relative humidity over time for the tests conducted in Fig 4c. The reproducibility of the experimental setup is demonstrated by the small standard deviations (below 6%) observed for both temperature and relative humidity for 3 separate testing as shown in Supplementary Fig. S12. Secondly, to address the reproducibility of the response of the tag, additional tests were performed with the three oils. The frequency response shift after the melting of the PCM is consistent across the three tested oils with shifts of 34.4 ± 2.8 MHz, 36.5 ± 5.3 MHz and 31.1 ± 4.4 MHz, for olive, jojoba and coconut oil respectively ($n = 3$), visible in the newly added Fig. S10. Those results, where the standard deviation of the frequency response is lower than 15%, show that the experiments are reproducible. The reproducibility of the experiments is further supported by the also lower than 15 % standard deviation of the response of the tags tested under inclination in Fig. 4d.

While the current manufacturing process relies on manual screen printing and the manual application of beeswax, the reproducibility of the tag, expressed as the deviation in frequency of resonance between samples, is only 1.6%, as noted in the manuscript (line 239). Finally, batch to batch improvements should be made in an industrialisation process to avoid calibration for each batch.

Fig. S10 | Confining the melted PCM over the tag using a plastic well leads to similar resonance frequency shifts for all three oils. The 144 μL oil cube melts as the temperature is increased from 0 $^{\circ}\text{C}$ to 30 $^{\circ}\text{C}$ with a plastic well used to confine the oil over the tag. The measured resonance frequency shifts for the olive, jojoba and coconut oils are presented on the boxplot on the right and are $34.4 \pm 2.8 \text{ MHz}$, $36.5 \pm 5.3 \text{ MHz}$ and $31.1 \pm 4.4 \text{ MHz}$, respectively ($n = 3$).

Fig. S12 | Average environmental parameters recorded over time once icepacks have been removed from the setup. The temperature and relative humidity data when exposing the tag to a gradual increase in temperature in experiments presented in Fig. 4 was averaged with shaded area representing the standard deviation ($n = 3$).

3. The manuscript should provide a comparison of the proposed technology's performance against currently available commercial products, focusing on key metrics such as sensitivity, accuracy, resolution, long-term stability, and response time.

Thank you for the insightful comment, which helped us better emphasize the novelty of this work. As suggested by you and other reviewers, we have added a comparison table summarizing other chipless tags designed for time-temperature excursion sensing. This **Table 1** is placed in the discussion section, as this location aligns well with the structure and flow of the manuscript. To avoid redundancy, we opted not to repeat the table in both the state-of-the-art section and the results section.

The **Table 1** includes a detailed comparison of key aspects such as materials, fabrication methods, frequency of operation, response time, stability and response. Material choice and fabrication method are important for sustainable development of sensors. The temperature threshold is of interest as various materials have been used in the Start-of-the-Art with different temperature transition points. The type of resonator and its operation frequency is added however the comparison is ambiguous as it is highly dependent on the shape of the chipless devices. The response, reproducibility and stability are essential parameters for sensing application, notably when using degradable materials. Some of the proposed metrics such as sensitivity or resolution were not considered as we do not believe they are appropriate for a sensor relying on an irreversible mechanism for detecting the crossing of a threshold temperature. As highlighted in the table, the proposed sensing chipless tag is the first of its kind to utilize degradable materials for the irreversible detection of temperature thresholds, offering a unique combination of eco-friendliness, reproducibility, and stability. Notably, this work is the only fully eco-friendly chipless printed sensor reported to date for irreversible detection of temperature thresholds. Unlike other systems, which are either partially eco-friendly or do not report long-term stability and reproducibility, the proposed chipless tag combines sustainability, reproducibility, and operational reliability.

Additionally, the ability to achieve temperature-controlled and irreversible detection, coupled with a detailed characterization of the RF performance of the system over time, highlights the novelty and scientific contribution of the proposed technology.

Table 1 | Summary of chipless resonators used for temperature-threshold crossing detection.

Reference	65	53	54	57	56	This work
Fabrication	Etched	Etched	Etched	Etched	Printed	Printed
Substrate (area, thickness)	Taconic TLX_0 (6 × 6 mm ² , 0.5 mm)	Taconic TLX_9 (20 × 20 mm ² , 2 mm)	Taconic TLX_8 (25 × 25 mm ² , 0.5 mm)	Taconic TLX_8 (36 × 36 mm, 0.5 mm)	Paper glued on blotting paper (30 × 30 mm ²)	PHBH-Cellulose (20 mm diam., 0.3 mm)
Conductor	Copper	Copper	Copper	Copper	Silver	Zinc
Sensing material	Phenanthrene (Sublimates)	NaCl-water (Uncontrolled melting)	Grapeseed & coconut oils (Uncontrolled melting)	Grapeseed, coconut & palm oils (Uncontrolled melting)	Grapeseed & coconut oils (Controlled melting)	Olive, jojoba & coconut oils (Controlled melting)
Degradable materials	No	Partially	Partially	Partially	Partially	Fully
Temperature threshold (°C)	72	-15 to -1	4, 8, 12	14.2, 17.3, 23.6	19.7, 21.9	8, 15, 25
Resonator	ELC	Ladder-shape (time-based)	Circular Asymmetric SRR	Circular Asymmetric SRR	Square SRR	ELC
Operating frequency (GHz)	6 to 7	3.5 to 5.5	3 to 6	3 to 5	2 to 4	1 to 5
Frequency shift (MHz)	320	NA	250	400-600	250, 600	30-40
Reproducible frequency shift	NA	NA	NA	NA	NA	Yes
Stability	NA	NA	NA	NA	NA	2 MHz drift after 1 year
Response time	60 min	NA	NA	NA	30 min	Few to 140 s*
Disintegration in compost	NA	NA	NA	NA	NA	Yes

* Depending on the temperature difference experienced by the tag.

Edits to the manuscript were done on lines 400 to 410 with the addition of the table.

4. What is the definition of “irreversible temperature changes”? In the respective context, irreversible changes refer to food spoilage rather than temperature alteration. Temperature is a parameter that is always reversible concerning changes in environmental conditions.

The terminology used was indeed inadequate. An irreversible temperature fuse is a device designed to provide a permanent indication of whether a specific temperature threshold has been exceeded. Irreversibility means the fuse cannot return to its original state once activated. The mention of “Irreversible temperature changes” in the abstract was replaced by the wording “to irreversibly track temperature changes through a permanent shift in resonance frequency”

Edit: Formulation changed in the abstract on lines 26 to 28 and in the introduction on lines 80, 85, 92 and 101.

5. Is ‘zinc metallic traces’ considered part of “exclusively biodegradable materials”?

Thank you for the judicious remark on the title of this manuscript.

Multiple publications have already referred to zinc as being a degradable, a bioresorbable or ecoresorbable material. To limit the confusion on the meaning of biodegradable in relation to the metallization, the word “exclusively” was removed from the title.

6. How significantly does the rate of temperature change impact this sensor? If there is a momentary spike in temperature, what type of response would the sensor exhibit?

To investigate the effect of a momentary spike in temperature on the response of the sensor, a new experiment was conducted. The effect of a spike of temperature (5 °C above the melting point of the PCM) on the melting time was investigated by considering different initial temperature for the frozen PCM. In this experiment, 50 μ L of frozen coconut oil was placed on a paper absorbent on the tag and stored at -10, 0, 10 or 20 °C. The tag was then abruptly exposed to 30 °C. The results show that as the difference in temperature between the tag with PCM and the environment decrease, so does the time to reach melting of the oil. Specifically, as seen in the newly added supplementary Fig. S13, when the initial temperature was 20 °C, the frequency of resonance of the tag started to shift after only 10 seconds of exposure to 30 °C while at an initial temperature of -10 °C, it took 140 seconds for the PCM to start melting. Therefore, if a sudden spike of temperature were to occur, the time for the PCM to melt would be based on its initial temperature with an amount of time that would decrease with an increase of the temperature spike.

Edits: Fig. S13 were added in supplementary and the results were commented in the manuscript from lines 349 to 356.

Fig. S13 | Time-temperature response of the tag with frozen coconut oil when exposed to air at 30 °C. a The frequency of resonance of the tag when suddenly exposed to 30 °C and 20% RH was measured over time for coconut oil in a frozen state at 4 initial temperatures of -10, 0, 10 and 20 °C. **b** The temperature and relative humidity of the air surrounding the tag is recorded during the various time-temperature experiment.

7. I understand that the author intends to utilize phase change materials (PCMs) for material modification due to elevated temperatures over specific durations, reflecting this through changes in the material's resonance frequency. However, traditional temperature sensors are more suited for long-term continuous temperature recording. In the current design, once the phase change of material happened, it no longer can restore to its original state.

Thank you for your comment. The goal of the presented chipless tag is to simply function as a one-time-use device, similar to a fuse, to avoid the use of silicon chips, battery, and enable compostability, cost-effectiveness with mass manufacturing by printing. In this publication, we did not intend to address continuous monitoring of temperature, as the phase change material (PCM) is designed to permanently respond to exceeding temperatures. Once this happens the tag is used and as pointed out; its initial state cannot be restored. If of interest and economically viable, the tag could be recovered, by cleaning it up and by implementing a new frozen oil puck and paper absorbent. Finally, in some logistic scenarios, multiple tags with different melting oils could be used to determine by how much the temperature went higher.

8. The author claim that the sensor can be used for sustainable IoT but only demonstrated its potential application in cold chain temperature monitoring. First of all, NFC communication has very limited communication distance, to what extend can this technology be useful for IoT? How close must the PCM be to the NFC reader to obtain an accurate measurement? Secondly, The author has only demonstrated initial results regarding the materials' performance under environmental temperature changes. However, the practical utility of the tag in real-world applications has not been adequately demonstrated.” Furthermore, what is the dimension of the sensor? Is the present design suitable to be integrated to cold chain temperature monitoring? Can the sensor be significantly miniaturized?

The current chipless sensor design operates based on close-proximity read-out. In cold chain monitoring, sensors are typically placed inside the parcel, near the object of interest. For items like frozen biological samples, the sensor may need to be manually extracted and read immediately upon arrival.

Regarding the practical utility of the sensor in real-world applications, we recognize that further testing and validation in diverse environments are required to fully demonstrate its potential. The custom-developed reader, which is priced similarly to a mid-range smartphone (under \$400), offers an affordable solution. The integrated nanoVNA, which facilitates S-parameter readout, costs around \$200 (significantly less than typical laboratory equipment) and the Beaglebone running the Python PCA script is priced under \$100. Although chipless technology requires a custom reader, the Wi-Fi functionality embedded in our custom reader allows seamless smartphone interaction, as shown in the accompanying video 2. This work thus provides a foundation for further exploration of chipless technology, including its application in cold chain monitoring.

As for the dimensions of the sensor, it is 22mm in diameter as seen in with the scale bar in **Fig. 2e**. However, its size is directly related to the resonance performance and operational frequency. Passive resonant structures (such as antennas and LC circuits) typically have dimensions proportional to the wavelength (e.g., quarter-wavelength or half-wavelength). As the frequency increases, the physical size of these structures decreases. Therefore, using higher frequencies can reduce the overall footprint of the sensor. While miniaturization is possible to a certain extent, there are limitations. Higher frequencies can help reduce the size of the resonant elements, but this comes with trade-offs in terms of fabrication complexity, efficiency, and sensitivity. In our case, a significant reduction in size is limited by screen printing accuracy, which restricts the size of the individual inductive and capacitive elements.

9. The author has stressed that the sensor is “chipless”. Please further elaborate this point by: (1) Explain what chips are used from conventional temperature sensor? And why the chipless design is preferred over the conventional design? (2) This present sensor design still applies NFC reader which contains chip to extract data, what is this sensor considered chipless?

Thank you for your valuable comment regarding the chipless design.

We appreciate the opportunity to clarify this issue. The term "NFC" was used inaccurately in the manuscript. While the tag operates in the near-field with the custom-made antenna, it does not communicate within the NFC frequency range (13.56 MHz) but rather in the microwave range (1-10 GHz). The tag does not rely on any NFC chips and is fully without silicon chips, i.e. chipless. Therefore, the terminology "NFC" has been replaced with "near-field electromagnetic coupling," and references to "NFC tag," "NFC reader," and "NFC antenna" have been corrected to "microwave RFID tag" and "microwave RFID reader antenna" throughout the manuscript. This revision ensures a more accurate representation of the chipless technology used in this work. The chipless design offers the advantage of being fully eco-friendly, as it does not contain any integrated silicon components or battery for operation.

We specifically addressed that the tag is chipless and while the reader is not, which is usual for this kind of RFID implementation. One can buy once a portable and performing reader to read the tags which are chipless and as we shown compostable.

Reviewer 2

Comments:

1. First of all, I believe two important aspects are missing and should be addressed: 1) A comparison of your developed system with other similar systems previously published, and based on that, 2) An explanation of what characteristics, or combination of characteristics, make your system unique and advance beyond the current state of the art.

Thank you for the insightful comment, which helped us better emphasize the novelty of this work. As suggested by you and other reviewers, we have added a comparison table summarizing other chipless tags designed for time-temperature excursion sensing. This table is placed in the discussion section, as this location aligns well with the structure and flow of the manuscript. To avoid redundancy, we opted not to repeat the table in both the state-of-the-art section and the results section.

The **Table 1** includes a detailed comparison of key aspects such as materials, fabrication methods, frequency of operation, response time, stability and response. Material choice and fabrication method are important for sustainable development of sensors. The temperature threshold is of interest as various materials have been used in the Start-of-the-Art with different temperature transition points. The type of resonator and its operation frequency is added however the comparison is ambiguous as it is highly dependent on the shape of the chipless devices. The response, reproducibility and stability are essential parameters for sensing application, notably when using degradable materials. As highlighted in the table, the proposed sensing chipless tag is the first of its kind to utilize degradable materials for the irreversible detection of temperature thresholds, offering a unique combination of eco-friendliness, reproducibility, and stability. Notably, this work is the only fully eco-friendly chipless printed sensor reported to date for irreversible detection of temperature thresholds. Unlike other systems, which are either partially eco-friendly or do not report long-term stability and reproducibility, the proposed chipless tag combines sustainability, reproducibility, and operational reliability.

Additionally, the ability to achieve temperature-controlled and irreversible detection, coupled with a detailed characterization of the RF performance of the system over time, highlights the novelty and scientific contribution of the proposed technology.

Table 1 | Summary of chipless resonators used for temperature-threshold crossing detection.

Reference	65	53	54	57	56	This work
Fabrication	Etched	Etched	Etched	Etched	Printed	Printed
Substrate (area, thickness)	Taconic TLX_0 (6 × 6 mm ² , 0.5 mm)	Taconic TLX_9 (20 × 20 mm ² , 2 mm)	Taconic TLX_8 (25 × 25 mm ² , 0.5 mm)	Taconic TLX_8 (36 × 36 mm, 0.5 mm)	Paper glued on blotting paper (30 × 30 mm ²)	PHBH-Cellulose (20 mm diam., 0.3 mm)
Conductor	Copper	Copper	Copper	Copper	Silver	Zinc
Sensing material	Phenanthrene (Sublimates)	NaCl-water (Uncontrolled melting)	Grapeseed & coconut oils (Uncontrolled melting)	Grapeseed, coconut & palm oils (Uncontrolled melting)	Grapeseed & coconut oils (Controlled melting)	Olive, jojoba & coconut oils (Controlled melting)
Degradable materials	No	Partially	Partially	Partially	Partially	Fully
Temperature threshold (°C)	72	-15 to -1	4, 8, 12	14.2, 17.3, 23.6	19.7, 21.9	8, 15, 25
Resonator	ELC	Ladder-shape (time-based)	Circular Asymmetric SRR	Circular Asymmetric SRR	Square SRR	ELC
Operating frequency (GHz)	6 to 7	3.5 to 5.5	3 to 6	3 to 5	2 to 4	1 to 5
Frequency shift (MHz)	320	NA	250	400-600	250, 600	30-40
Reproducible frequency shift	NA	NA	NA	NA	NA	Yes
Stability	NA	NA	NA	NA	NA	2 MHz drift after 1 year
Response time	60 min	NA	NA	NA	30 min	Few to 140 s*
Disintegration in compost	NA	NA	NA	NA	NA	Yes

* Depending on the temperature difference experienced by the tag.

Edits to the manuscript were done on lines 400 to 410 with the addition of the table.

2. Although the use of biodegradable materials is a clear advantage from a sustainability perspective, it also limits the operational lifespan of the tag. In fact, the printed zinc affects the performance of the RLC circuit as early as week 1. Therefore, how long can the tag remain functional before becoming ineffective due to biodegradation? This limitation significantly restricts the potential applications of the system in real-world scenarios and should be clearly addressed in the manuscript.

The degradation of the tag is not expected to occur during its intended use and was demonstrated to be stable in the operative environment. The composting experiment was conducted to evaluate the disintegration of the tag at the end of its life, and no wireless measurements were intended to be performed during the decomposition process.

To address concerns regarding the stability of the tag, shelf-life was analysed under ambient temperature and humidity conditions. The evolution of the resonance frequency for a beeswax encapsulated tag was measured after one year. The resonance frequency exhibited only a minor change of 2 MHz and an amplitude drop of 16 dB. The resonance frequency shift due to aging is smaller than the 30 MHz induced frequency shifts and demonstrates that the encapsulated tag with beeswax can remain functional over an extended period.

Edits to the manuscript: Stability after 1 year was addressed from lines 239 to 246 with the addition of supplementary Fig. S7.

Fig. S7 | Stability of the printed chipless tag after 1 year at ambient conditions. The frequency of resonance of the tag with beeswax, after fabrication and 1 year later. The frequency of resonance of the encapsulated tags exhibits a drift of only 2 MHz after 1 year with an amplitude drop of 16 dB.

3. One aspect that has particularly caught my attention is the incorrect conceptual use of Near Field Communication (NFC) technology in this work. The authors frequently refer to NFC communication, including terms such as "chipless NFC tag," "NFC reader," and "NFC antenna." However, this is inaccurate. Firstly, NFC technology operates at a frequency of 13.56 MHz, whereas the developed tag functions in the GHz range. Secondly, NFC is a set of communication protocols, which makes the term "chipless NFC tag" fundamentally incorrect, as these protocols require a chip to facilitate communication. For example, the communication between an NFC tag and an NFC-enabled smartphone is made possible through these protocols. It is therefore misleading to state that the authors have developed a custom NFC antenna or reader. Additionally, it is quite surprising that the custom reader purportedly detects and reads the chipless tag, yet then uses an entirely different wireless protocol (WiFi) to transmit the data to the smartphone, as demonstrated in the supplementary video. Clearly, NFC is not the technology being employed in this work. I kindly request that this significant conceptual error be addressed and corrected.

Thank you for your valuable comment regarding the incorrect conceptual use of NFC technology.

We appreciate the opportunity to correct this issue. As you judiciously pointed out, the term "NFC" was used inaccurately in the manuscript. While the tag operates in the near-field, it does not communicate within the NFC frequency range (13.56 MHz) but rather in the microwave range (1-10 GHz). Therefore, the terminology "NFC" has been replaced with "near-field electromagnetic coupling," and references to "NFC tag," "NFC reader," and "NFC antenna" have been corrected to "microwave RFID tag" and "microwave RFID reader" throughout the manuscript. This revision ensures now a more accurate representation of the chipless technology used in this work.

4. I was also surprised by the use of Machine Learning by the reader to differentiate between the tags. Is it truly necessary to employ ML for identifying only three different resonance frequencies? It seems excessive for such a purpose.

The term "machine learning" was initially used as a broad descriptor. However, since the sensor is manually manufactured, and small variations in the reproducibility of the tags are possible, the detection and classification of the passive tags were carried out using a data analysis approach based on principal component analysis (PCA) as the core method.

Python implementations were developed to support consecutive numerical identifiers, with histograms stored as Python lists, and arbitrary identifiers, with histograms stored as dictionaries. The Python script runs on the Beaglebone embedded in the custom-made reader.

The PCA method enables the separation and identification of various tag states, as demonstrated in the accompanying video.

Edits: The terminology "machine learning" has been replaced with "PCA method" throughout the manuscript, and additional details regarding this approach have been included in the methods section from line 547 to 553.

5. Although the concept is ingenious, the need for results to be obtained using bulky and expensive laboratory equipment (e.g., a VNA) or a custom-developed reader significantly limits its potential application in real-world scenarios, such as smart packaging. While the authors acknowledge this limitation in the Discussion section, I believe it poses a substantial constraint that could reduce the likelihood of publication in very high-impact journals.

Thank you for your comment regarding the applicability of the technology

In terms of cost, the custom-developed reader is comparable to a mid-range smartphone, with a total cost of under \$400. Specifically, the integrated nanoVNA, which facilitates the S-parameter readout, is priced at approximately \$200 at the time of writing—considerably more affordable than traditional laboratory equipment. Additionally, the Beaglebone used to run the Python PCA script costs less than \$100.

We acknowledge that chipless technology inherently requires a custom reader for operation. However, we have designed our custom reader to enhance accessibility and practicality. For instance, the reader incorporates embedded Wi-Fi functionality, enabling seamless interaction with smartphones—a feature commonly utilized in logistics and smart packaging applications. This functionality is demonstrated in the accompanying video 2.

To further address your concern, we have expanded the discussion section to emphasize the cost-effectiveness and practicality of the custom reader, particularly its suitability for real-world applications such as smart packaging. These updates have been included in the discussion on lines 420 to 431.

Reviewer 3

Comments:

1. State of the Art in Chipless RFID: Include a review of similar research works and publications to highlight the novelty and scientific contribution of the proposed work.

Thank you for the insightful comment, which helped us better emphasize the novelty of this work. As suggested by you and other reviewers, we have added a comparison table summarizing other chipless tags designed for time-temperature excursion sensing. This table is placed in the discussion section, as this location aligns well with the structure and flow of the manuscript. To avoid redundancy, we opted not to repeat the table in both the state-of-the-art section and the results section.

The **Table 1** includes a detailed comparison of key aspects such as materials, fabrication methods, frequency of operation, response time, stability and response. Material choice and fabrication method are important for sustainable development of sensors. The temperature threshold is of interest as various materials have been used in the Start-of-the-Art with different temperature transition points. The type of resonator and its operation frequency is added however the comparison is ambiguous as it is highly dependent on the shape of the chipless devices. The response, reproducibility and stability are essential parameters for sensing application, notably when using degradable materials. As highlighted in the table, the proposed sensing chipless tag is the first of its kind to utilize degradable materials for the irreversible detection of temperature thresholds, offering a unique combination of eco-friendliness, reproducibility, and stability. Notably, this work is the only fully eco-friendly chipless printed sensor reported to date for irreversible detection of temperature thresholds. Unlike other systems, which are either partially eco-friendly or do not report long-term stability and reproducibility, the proposed chipless tag combines sustainability, reproducibility, and operational reliability.

Additionally, the ability to achieve temperature-controlled and irreversible detection, coupled with a detailed characterization of the RF performance of the system over time, highlights the novelty and scientific contribution of the proposed technology.

Table 1 | Summary of chipless resonators used for temperature-threshold crossing detection.

Reference	65	53	54	57	56	This work
Fabrication	Etched	Etched	Etched	Etched	Printed	Printed
Substrate (area, thickness)	Taconic TLX_0 (6 × 6 mm ² , 0.5 mm)	Taconic TLX_9 (20 × 20 mm ² , 2 mm)	Taconic TLX_8 (25 × 25 mm ² , 0.5 mm)	Taconic TLX_8 (36 × 36 mm, 0.5 mm)	Paper glued on blotting paper (30 × 30 mm ²)	PHBH-Cellulose (20 mm diam., 0.3 mm)
Conductor	Copper	Copper	Copper	Copper	Silver	Zinc
Sensing material	Phenanthrene (Sublimates)	NaCl-water (Uncontrolled melting)	Grapeseed & coconut oils (Uncontrolled melting)	Grapeseed, coconut & palm oils (Uncontrolled melting)	Grapeseed & coconut oils (Controlled melting)	Olive, jojoba & coconut oils (Controlled melting)
Degradable materials	No	Partially	Partially	Partially	Partially	Fully
Temperature threshold (°C)	72	-15 to -1	4, 8, 12	14.2, 17.3, 23.6	19.7, 21.9	8, 15, 25
Resonator	ELC	Ladder-shape (time-based)	Circular Asymmetric SRR	Circular Asymmetric SRR	Square SRR	ELC
Operating frequency (GHz)	6 to 7	3.5 to 5.5	3 to 6	3 to 5	2 to 4	1 to 5
Frequency shift (MHz)	320	NA	250	400-600	250, 600	30-40
Reproducible frequency shift	NA	NA	NA	NA	NA	Yes
Stability	NA	NA	NA	NA	NA	2 MHz drift after 1 year
Response time	60 min	NA	NA	NA	30 min	Few to 140 s*
Disintegration in compost	NA	NA	NA	NA	NA	Yes

* Depending on the temperature difference experienced by the tag.

Edits to the manuscript were done on lines 400 to 410 with the addition of the table.

2. Fig. 1 Clarification: Ensure that the papers in Fig. 1 are of the same thickness as the 200-micron PHBH/Cellulose paper, as variations in thickness can significantly impact absorption and S11 response.

Thank you for your remark, which has helped improve the clarity of the manuscript. All tested materials were compared at a thickness of 200 microns \pm 15%. The manufactured substrates, including the PHBH-based substrates, were 200 μ m-thick and the printed electronics (PE) paper, was available at a thickness of 230 μ m. While the PE paper was only 15% thicker, its weight-increase at 70% RH compared to the PHBH-cellulose sample was double, as illustrated in Fig. 1c.

In Fig. 1e, the results demonstrate that the PHBH-cellulose substrate exhibits less than a 0.3% change in resonance frequency as relative humidity (RH) increases. While the resonance at 30% RH for the PHBH-based microstrip line is slightly shifted compared to the paper-based microstrip line (due to differences in dielectric properties and the 30 μ m thinner substrate) the PHBH-cellulose substrate shows less than a 0.3% resonance frequency shift at 70% RH, in contrast to the 6.6% shift observed with the PE paper.

The manuscript was updated, given the thicknesses of the materials in the caption of the Fig. 1c and 1e.

3. Water Vapor Sorption Test: Clarify why 30-70% RH was selected and how it correlates to RH in a cold chain environment. The manuscript should also specify the temperature used in the test and its relevance to actual cold supply chain conditions.

Thank you for your comment regarding the 30-70% RH range and temperature used in the water vapor sorption test.

The purpose of this test was to compare the substrates' materials based on their properties, rather than assess performance in a specific logistics environment. The 30-70% RH range was chosen to highlight differences in material behaviour. While not simulating cold chain conditions, this test characterizes the moisture sensitivity of various substrates.

The testing at 40 °C (shown in Fig. 1c) was selected because it increases absolute humidity compared to room temperature, offering a more stringent scenario for evaluating materials under elevated humidity.

4. RF Measurement Conditions: Provide details on the temperature during RF measurements and the relationship between temperature and RH. Experimenting with the impact of temperature while keeping RH constant is strongly suggested.

Thank you for the suggestion to improve the result sections.

In our previous publication (<https://doi.org/10.1002/aelm.202400229>), the resonance frequency of a zinc-printed microstrip line on paper was studied. The study demonstrated that the thermal coefficient of resistivity of zinc influenced the resonance frequency under constant relative humidity. In the context of the present study, we considered whether this thermal shift could affect the resonance of the chipless tag with oil, as depicted in Fig. 3C. Our data show that the thermal effect on the resonance frequency of the tag before the oil melting is less than 10 MHz. In contrast, the oil melting induces a further frequency shift of more than 30 MHz. It is important to note that the direction of the thermal shift is consistent with the induced frequency shift caused by melting. Therefore, this would not pose a problem for the detection of the irreversible resonance frequency shift.

Regarding the cold chamber testing and oil melting experiments, the temperature and relative humidity data were averaged across tests and are presented in Supplementary Fig. S12. These data indicate good reproducibility of the experiment ($n = 3$). Due to the experimental setup, where ice packs were used to reduce the temperature in the chamber, the relative humidity could not be tightly controlled. As shown in Supplementary Fig. S12, high relative humidity (90% RH) occurred when the chamber temperature was below 10 °C, due to condensation of air after ice pack removal. Once the temperature reached 10 °C, the chamber air progressively dried, stabilizing at an average relative humidity of 30%. Despite this, the low standard deviation (indicated by the shaded area) demonstrates the consistency of the experimental results.

Furthermore, the data suggest that humidity had a negligible effect on the response of the tag. As the air dried, the value of the capacitor in the LC circuit would have been expected to decrease due to the reduction in relative humidity. However, the observed response of the tag was opposite to this expectation, indicating that the frequency shift was primarily driven by temperature changes and the oil melting process.

To further investigate the time-dependent response of the tag when exposed to temperatures above the melting point of the phase-change materials (PCMs), we conducted an additional experiment, keeping the relative humidity (RH) constant at 20%, as requested. Tags with coconut oil, initially stored at -10, 0, 10, and 20 °C, were abruptly exposed to a constant temperature of 30 °C (approximately 5 °C above

the melting point of coconut oil). The resonance frequency was recorded every second as the oil melted into the capillary absorbent. The results are presented in Supplementary Fig. S13. As expected, the response of the tag was faster when its initial temperature was closer to the melting point of the oil. For example, the tag initially at 20 °C began showing signs of melting in less than 10 seconds, whereas the tag initially at -10 °C took over 2 minutes to exhibit similar signs.

Finally, regarding Fig. 1d and Fig. 1e, we have clarified that the temperature during these experiments was 25 °C

Edits to the Manuscript:

The temperature values during testing for the results presented in Fig. 1 were included in the caption.

Temperature and humidity values during the cold chamber tests were added as Supplementary Fig. S12 and discussed in the revised manuscript (lines 349–352).

A new supplementary figure (Fig. S13) showing the time-temperature response of various tags exposed to 30 °C was added and is discussed in lines 352–355.

Fig. S12 | Average environmental parameters recorded over time once icepacks have been removed from the setup. The temperature and relative humidity data when exposing the tag to a gradual increase in temperature in experiments presented in **Fig. 4** was averaged with shaded area representing the standard deviation ($n = 3$).

Fig. S13 | Time-temperature response of the tag with frozen coconut oil when exposed to air at 30 °C. **a** The frequency of resonance of the tag when suddenly exposed to 30 °C and 20% RH was measured over time for coconut oil in a frozen state at 4 initial temperatures of -10, 0, 10 and 20 °C. **b** The temperature and relative humidity of the air surrounding the tag is recorded during the various time-temperature experiment.

5. Decomposition Rate: Calculate the decomposition rate considering the initial mass of the paper, as larger or thicker paper will take longer to decompose.

Including the decomposition rate would have enhanced the understanding of the degradation process. However, it was not added to the manuscript because the PHBH-based samples accumulated compost particles during the experiment, as evidenced by the dark particles visible in the images in Fig. 1f. This accumulation prevented accurate mass measurements for both substrates.

Nevertheless, the observation that the composite decomposes faster remains valid. The initial masses of the pure PHBH and the PHBH composite were 160.1 mg and 173.0 mg, respectively. Additionally, the PHBH composite substrate, which degraded first, was 3.8% thicker than the pure PHBH, with both substrates having the same surface area in contact with the compost.

Edits: Additional information on the initial mass were added to the manuscript in the caption of Fig. 1f and on lines 148 from 151.

6. Chipless Tag Design and Test Setup: Justify the change in thickness from 200 to 304 microns and provide experimental support for the claim that 304 microns minimises undesirable bending. Further tests are needed to determine the thickness threshold for minimal bending.

Thank you for your comment regarding the use of the term "bending." The wording has been revised to refer specifically to the "warping" of the tag.

For the 200 μm -thin substrate, warping was observed following the photonic sintering process, as the heat generated during sintering caused deformation of the substrate. To address this issue, the thickness of the PHBH-cellulose substrate was increased by 50%, resulting in a 300 μm substrate. This modification effectively prevented warping after the sintering step, as shown in the newly included Supplementary Fig. S4.

Reducing the energy delivered during the photonic sintering process could be another approach to avoid warping of the 200 μm PHBH-cellulose substrate. However, this would result in lower conductivity of the zinc track, as previously demonstrated in the literature on paper substrates

(<https://doi.org/10.1016/j.mne.2023.100218>). Such a reduction in conductivity would negatively impact RF performance by decreasing the quality factor (Q), where $Q = \frac{1}{R} \sqrt{\frac{L}{C}}$.

Regarding the bending threshold, mechanical failure tests were done when bending the tag over various radii of curvature and results are detailed in the next comment.

Edits: Lines 234 to 236 were updated to give more information on the change of thickness to minimize warping and reference to the supplementary Fig. S4 was added.

Fig. S4 | Warping of the 200 μm-thick substrates after photonic sintering of the zinc layer as opposed to the 300 μm-thick substrates. Optical images showing 4 different tags after photonic sintering of the zinc printed layer. On the left, wrinkles due to warping can be seen on the front and back side of 200 μm-thick devices. On the right, the 300 μm-thick tags do not show any signs of warping after sintering.

7. Impact of Substrate Deformation: Address how the proposed RFID sensor minimises the impact of mechanical deformation on frequency shifts.

To enhance the clarity of the results, a bending test was conducted to determine the maximum radius of curvature the tag could withstand before experiencing mechanical failure.

With a thickness of 300 μm, the tag demonstrates sufficient rigidity to remain flat even when bent to a radius of curvature as small as 10 mm. Furthermore, the coin-like shape of the tag inherently limits bending by discouraging favourable orientations for significant deformation.

The PHBH-cellulose substrate retains its flatness up to a radius of curvature of 5 mm, at which it breaks as shown in the newly added supplementary Fig. S5. Importantly, the RF performance of the tag was only slightly affected with a frequency change of 3 MHz when bent to a 20, 15 or 10 mm radius of curvature, as illustrated in the added Fig. S6. These radii are significantly more extreme than those typically encountered on standard cardboard packaging.

Edits:

Two supplementary figures were added. Fig. S5 showing the tag over various angles of curvature and its resulting mechanical deformation as well as Fig. S6 showing the tag S_{11} response before and after bending at 10 mm radius of curvature was added and commented on lines 239 to 241.

Fig. S5 | Bending of the tag with and without beeswax encapsulation over different radii of curvature. Optical images showing the tag, before, during and after being bent over structures having 20, 15, 10 and 5 mm radii of curvature. The 300 μm -thick tag breaks when bent over a 5 mm radius, regardless of the presence of beeswax. The 100 μm -thick beeswax encapsulation layer prevented the tag from breaking into two pieces, despite the failure of the PHBH-composite substrate.

Fig. S6 | S_{11} response of the tag with beeswax encapsulation before and after bending at 20, 15 and 10 mm radius of curvature. The frequency of resonance of the pristine tag after bending shifted by only 3 MHz for all cases.

8. NFC Antenna Usage: Clarify the discrepancy between using an NFC antenna for RF measurement, given the different operating frequencies.

Thank you for your insightful comment regarding the incorrect conceptual use of NFC technology.

As previously addressed while answering reviewers 1 and 2, we appreciate the opportunity to clarify this issue. You are absolutely correct that the term "NFC" was inaccurately used in the manuscript. Although the tag operates in the near-field, it does not utilize the NFC frequency range (13.56 MHz) but instead operates in the microwave range (1–10 GHz). To address this, references to "NFC tag," "NFC reader," and "NFC antenna" have been revised to "microwave near-field tag" and "microwave near-field reader antenna" throughout the manuscript. These revisions ensure a more precise and accurate representation of the chipless technology utilized in this work. The wording of reader and reader antenna has also been made consistent

9. RF Measurement Setup: Improve the RF measurement setup to follow standard practices, such as using an anechoic chamber to minimise interference and provide reliable results.

The tag was tested in a cold chamber to ensure precise temperature transitions during sensing. Incorporating an anechoic chamber into the current setup was deemed impractical, as it would compromise the necessary temperature control required for the experiment.

Nevertheless, due to the near-field operation of the system and the distance of 2 mm between the reader and the tag, RF disturbances from the surrounding environment are expected to have a negligible impact. Additionally, the primary goal of this sensor tag is to deliver a robust sensing solution capable of reliable operation in realistic environments, which may include clutter and interference. While measurements in an anechoic chamber could provide controlled RF characterization, they would not accurately reflect the sensor's intended operating conditions or its performance in practical applications.

10. Oil Frequency Shift Analysis: Explain why olive oil induces the largest frequency shift compared to other oils and provide a critical analysis of this observation.

For all tested oils, tags containing 144 μL oil blocks were placed on the reader within a cold chamber set to 0 °C, which was then heated to 30 °C. Among the oils, olive oil exhibits the lowest transition temperature and therefore melts first. As it melts, it spreads across a larger surface area of the tag during the time required for the chamber to reach 30 °C. In contrast, coconut oil and jojoba oil, with higher melting points, have less time to spread before reaching 30 °C.

When olive oil melts, it induces a frequency shift that stabilizes at an initial plateau. However, as the temperature continues to rise, the oil spreads beyond the edges of the tag and onto the reader surface, as shown in the newly added Supplementary Fig. S9. This leakage modifies the dielectric properties of the tag, resulting in undesired behaviour. To address this issue, a paper absorbent was introduced to confine the oil and prevent leakage beyond the tag area. To further investigate this phenomenon, a new experiment was conducted to demonstrate that the frequency shift remains consistent across all oils when the melted liquid is confined to a specific area. For this experiment, a plastic circular well was placed over the tag (without the paper absorbent) to contain the melted oil, as shown in Supplementary Fig. S10. The frequency shift induced by the melting of 144 μL blocks of olive, jojoba, and coconut oil was calculated and found to be similar across all oils when the liquid remained confined.

Edits: Two supplementary figures have been added. Fig. S9 illustrates the spreading of olive oil beyond the tag, while Figure S10 shows the corresponding frequency shift, averaged over three sensors, for all three oils when the phase change material (PCM) melts. Additionally, further details have been incorporated into the manuscript to describe these observations, added between lines 303 and 307.

Fig. S9 | Evolution of the spreading of the olive oil over time. The 144 μL olive oil cube melts as the temperature is increased from 0 $^{\circ}\text{C}$ to 30 $^{\circ}\text{C}$. At higher temperatures, when the liquid oil reaches the edge of the tag, it can be seen seeping between the tag and the reader antenna. This results in undesired modification of the dielectric properties of the tag and explains the second drop in frequency of resonance of the olive oil-based tag in **Fig. 3f**.

Fig. S10 | Confining the melted PCM over the tag using a plastic well leads to similar resonance frequency shifts for all three oils. The 144 μL oil cube melts as the temperature is increased from 0 $^{\circ}\text{C}$ to 30 $^{\circ}\text{C}$ with a plastic well used to confine the oil over the tag. The measured resonance frequency shifts for the olive, jojoba and coconut are presented on the boxplot the right and are $34.4 \pm 2.8 \text{ MHz}$, $36.5 \pm 5.3 \text{ MHz}$ and $31.1 \pm 4.4 \text{ MHz}$, respectively ($n = 3$).

11. Beeswax Impact: Test and report whether beeswax affects RF performance and its mechanical properties.

Absorbance Test: Include the relationship among the absorbance of the filter paper, its mass, the volume of oil, and the contact area of oil.

The beeswax had no significant impact on the mechanical properties of the tag, as shown in the newly added supplementary Fig. S5. Regardless of the presence of beeswax, the minimum bending radius under ambient conditions was found to be between 5 and 10 mm. However, beeswax does influence the maximum operational temperature, as this encapsulation melts at 62 $^{\circ}\text{C}$, but would not cause a problem in a cold chain environment. As demonstrated in the new Fig. S7, the beeswax enhances the RF stability of the tag, ensuring that the resonance frequency remains even after 1 year under ambient conditions despite the tag being made of degradable materials.

Fig. S7 | Stability of the printed chipless tag after 1 year at ambient conditions. The frequency of resonance of the tag with beeswax, after fabrication and 1 year later. The frequency of resonance of the encapsulated tags exhibits a drift of only 2 MHz after 1 year with an amplitude drop of 16 dB.

The addition of a 100 μm -thick beeswax encapsulation shifts the resonance frequency of the printed tag by 80 MHz and decreases its amplitude, as the increased dielectric loss, induced by the beeswax, impacts the RF profile of the device, as shown below:

However, since the tag needs to be encapsulated with beeswax to prevent oxidation of the zinc the tag is only intended to be used with the encapsulation. The change in the RF behaviour of the tag has no implication on the performance of the tag when PCM melts as shown by the obtained results. Therefore, this figure was not added in supplementary.

As already shown in Fig. S11, to know what is the maximum volume of oil absorbed by the capillary element and select the best candidate, a study was conducted by weighting various cellulose samples of known areas once fully saturated with oil. The volume was extrapolated using the density of jojoba oil of 0.87g/mL. The relation for the Whatman GB003 cellulose absorbent is:

$$\text{volume absorbed } (\mu\text{L}) = 0.54 * \text{area of paper } (\text{mm}^2)$$

Considering the density of jojoba, we get,

$$\text{Maximum oil mass absorbed (mg)} = \frac{0.54}{0.87} * \text{area of paper (mm}^2\text{)}$$

This relation was added in the caption of supplementary Fig. 16:

Fig. S11 | Absorption capacity of commercially available cellulose filter elements based on surface area. Liquid jojoba oil at room temperature is soaked in laser cut paper filters disk ranging from 20 mm² to 314 mm² in size. Maximum volume absorbed is calculated using the density of jojoba oil (0.87 g/mL). We have the following equation for the selected 100% cellulose Whatman GB003 absorbent:

$$\text{Maximum oil mass absorbed [mg]} = \frac{0.54}{0.87} * \text{area of paper [mm}^2\text{]}$$

12. Machine Learning Justification: Justify the use of machine learning for a simple differentiation task and provide technical details about the approach. If this section does not contribute to the paper, consider removing it.

As addressed in a previous answer to reviewer 2:

The term "machine learning" was initially employed as a general descriptor. However, since the sensor is manually fabricated and small variations in tag reproducibility are possible, the detection and classification of the passive tags were performed using a data analysis approach centered on principal component analysis (PCA).

To support this process, Python implementations were developed to handle consecutive numerical identifiers, with histograms stored as Python lists, and arbitrary identifiers, with histograms stored as dictionaries. The Python script operates on the BeagleBone embedded within the custom-made reader. The PCA method effectively enables the separation and identification of various tag states, as demonstrated in the accompanying video.

Edits have been made to address this clarification. The term "machine learning" has been replaced with "principal component analysis" throughout the manuscript, and further details on the PCA-based approach have been incorporated into the methods section (lines 548–553).

13. Supplementary Figure 7a: Clarify why both pure PHBH and 50% PHBH/cellulose materials have a trough around 50°C in the DSC graph.

The trough around 50 °C corresponds to the cold exothermal crystallisation temperature of the PHBH. This crystallisation is due to the alignment of the polymer chains due to the material softening above the glass transition point. This trough is similar to those found in other scientific publication of PHBH.

([10.1098/rsos.211485](https://doi.org/10.1098/rsos.211485) and <https://www.sciencedirect.com/science/article/pii/S2589234723000659>)

Edits: Explanation of the trough was added to the manuscript from lines 169 to 172 with two additional references (60 and 61). The supplementary Fig. 7a was renamed to Fig. S2a

Dear Reviewers,

Thank you for the second revision of the manuscript. Following new high-frequency simulations of the tag and antenna and bending experiments, we believed to have answered your concerns and have improved the quality of the manuscript.

Firstly, we have answered the comments of reviewer #1. To prevent confusion, the terms “sensing” and “sensor” were replaced with “thermal indicator” and “temperature responsive tag”. The variability of the response of the tag was explained and mentioned with more clarity in the manuscript. The biodegradability nature of the zinc was strengthened with the addition of more scientific references. The feasibility of the near-field operation and the characterization of the coupling between the tag and the reader antenna were addressed.

Secondly, we thank reviewer #2 for his/her comments that help improve the manuscript.

Lastly, to answer the concerns of reviewer #3, we conducted additional bending tests and measured the response after cycling. A detailed Ansys HFSS simulation of the reader antenna and its coupling with the tag was conducted to determine the radiation patterns. The table 1 was updated to better compare the presented work with existing chipless temperature-responsive devices.

All changes have been highlighted in the manuscript and supplementary information, and a detailed point-by-point response to your comments is included below.

Answer to reviewers:

Reviewer #1 (Remarks to the Author):

Thank you for addressing my previous comments and revising the manuscript. The authors have made commendable efforts to improve the clarity and scope of the work. However, several critical issues remain unresolved and require further attention to ensure the manuscript meets the high standards of Nature Communications. Below are my detailed concerns and recommendations:

1. Terminology: “Temperature Sensor” is Misleading

The manuscript continues to describe the device as a “temperature sensor.” However, the system does not fulfill the technical definition of a sensor, which is expected to reversibly quantify thermal energy and convert it into a measurable signal. Instead, the presented device operates irreversibly via phase-change material (PCM) properties. To avoid misinterpretation, I strongly recommend replacing “sensor” with a more accurate term such as “temperature-responsive tag,” “thermal indicator,” or “PCM-based monitor” throughout the manuscript. This revision is critical to maintain scientific rigor and prevent confusion among readers.

Thank you for your comment on the confusion induced by the misleading terminology used. The concept of temperature sensor was replaced across the manuscript with “thermal indicator” and “temperature-responsive”. To address the idea of crossing of temperature, the terminology “temperature-excursion” is now used. The “sensor tag” is now referred solely as “tag” to limit confusion on the irreversibility aspect of the device.

2. Statistical Significance of Frequency Response Variability

The authors note a 15% standard deviation in the frequency response across repeated tests (as stated in the response letter). While this provides some insight into reproducibility, it is suggested that the manuscript:

- **Explicitly include this value in the Results/Discussion section.**
- **Justify whether 15% variability is acceptable for the intended application (e.g., by benchmarking against industry standards or comparable studies).**
- **Discuss potential sources of variability (e.g., fabrication tolerances, environmental factors) and strategies to mitigate them.**

The largest 15% standard deviation that was included in the response letter was explicitly added while discussing Fig. 4 on lines 363 and 364: “The largest standard deviation in the frequency shift after melting for the different inclinations was measured to be 15% of the response.”

Other studies evaluating the temperature-response using chipless tags (presented in Table 1) have not included standard deviation in their frequency shift response making benchmarking difficult. The 15% variability of the total response after melting, corresponding in the worst case to a deviation of 5 MHz for a 30 MHz response as shown in Fig. 4, is deemed to be acceptable. The variability does not affect the system performance as it operates in the GHz with limits of detection of less than 1 MHz and the detection of temperature-crossing was shown to also be robust at different inclinations. Furthermore, any potential impact due to deviation can be corrected by the PCA algorithm behind the read-out system that would be applied in a real-life scenario. Lines 390-392 were added with the following: “Furthermore, any potential impact due to the variability of the response after melting, measured to be at most 5MHz or 15%, could be corrected by the PCA algorithm”

As mentioned in the manuscript, the environmental parameters during experimental testing were highly reproducible, therefore variability is expected to be device specific. Variability can arise from different batches in the PHBH-cellulose substrate fabrication and its final thickness. As the tag is laboratory manufactured, the printing of the zinc layer and its sintering or the beeswax coating thickness could have an effect in the variability across samples. Finally, the way the PCM melts and saturates the paper absorbent could be a reason of variability. Nevertheless, we deemed the variability acceptable given the lab-scale manufacturing methods. The integration of automation in the fabrication process, namely printing and encapsulation, could help improve on the variability across samples.

The following comment was added in the manuscript lines 364 to 369: “The variability in the response is suspected to be due to manual fabrication of the tag, more precisely the PHBH-cellulose substrate batch fabrication, its thickness, the printing of the zinc and the beeswax encapsulation. The variability in the response can also be due to the way the PCM melts and saturates the paper absorbent. The standard deviation is expected to decrease by integrating process automation in the manufacturing of the chipless tag.”

3. Citations for Zinc as a Bioresorbable Material

The use of zinc as a degradable material is central to the device’s design. However, the manuscript lacks references supporting zinc’s bioresorbability and degradation mechanisms. Relevant citations should be added to strengthen this claim and contextualize the material choice.

We agree that supporting references are essential to substantiate the choice of zinc as a bioresorbable material. In response, we have added the references 40 to 46 to the revised manuscript to support the established zinc bioresorbability and its degradation mechanisms. We have also updated the manuscript with the following sentence from lines 72 to 75: “The zinc bioresorbability and degradation mechanisms via electrochemical corrosion, forming biocompatible zinc ions and oxides, make it particularly attractive for transient electronic applications⁴⁰⁻⁴⁶” with added reference 40 to 46.

4. Practical Limitations of Response Time and Economic Viability

The authors’ new experiments reveal that the system’s response speed depends on the PCM’s initial storage temperature. While this is an important finding, the manuscript is suggested to address:

- **Applicability Concerns: How does the response time align with real-world scenarios (e.g., food spoilage timelines)? If the tag’s response is slower than the spoilage process, it may fail as a safety indicator. Conversely, if it responds too quickly, it risks premature inactivation.**
- **Economic/Technical Comparison: How does this system compare economically and functionally to existing temperature sensors (e.g., cost, ease of deployment, reusability)? A brief discussion of trade-offs would clarify the technology’s niche.**

Compared to conventional temperature sensors, our chipless system offers a low-cost, battery and chip free sustainable alternative suitable for disposable large-scale deployment. While traditional sensors may provide continuous monitoring and reusability, they often require power sources and higher integration complexity. In contrast, our chipless tag system is designed for single-use, sustainable and cost-sensitive packaging applications, acting as a first-line indicator of temperature exceedance.

Our system is intended for simple use cases where a one-time indication of a temperature excursion (from T1 to T2) is sufficient to trigger a quality check. It does not support continuous monitoring or record the exact timing of the event but instead alerts on the fact that the temperature of a product has exceeded a critical temperature. For applications requiring time–temperature tracking, the system could be adapted implementing PCMs engineered for specific thermal profiles.

We have included this mention in the result section of the manuscript on lines 375 to 377 with newly added reference 72 to 74, addressing the cold chain time constraints: “The system is therefore fast enough to detect a temperature breach before safety thresholds are exceeded^{72–74}. If finer time-dependence is required, PCM and absorbent materials can be adjusted accordingly.”

In the discussion, the following text was added from lines 428 to 435: “The current version of the tag serves as a first-line indicator for cold chain breaches. It offers a simple, one-time alert that can help identify if a temperature excursion took place and where along the logistics chain. This initial indication can prompt further investigation to assess whether the goods were affected and to determine the cause of the breach. Eventually, a fresh chipless tag could be deployed for further tracking of the goods during the remainder of their transport. For applications where time–temperature tracking is of interest, the system could be redesigned with tailored PCMs with specific thermal profiles”

And finally, the trade-offs are now as well addressed in the discussion from lines 443 to 451: “This system is designed for applications in the perishable goods supply chain as a simple one-time indicator of temperature-exceedance in the cold chain. Our sustainable approach provides low-cost, battery-free deployment at scale in comparison to conventional solutions that offer continuous monitoring and data logging but require embedded power supply and more consequent integration. Moreover, looking specifically at active and passive RFID devices, our chipless design removes the need for silicon-based circuitry, reducing both cost and environmental impact, and making it better suited for disposable and high-volume applications. Compared to colorimetric indicators, it offers desired performance for similar cost and easy integration in packaging while avoiding optical sighting issues like subjective interpretation.”

5. Feasibility of Near-Field Operation (2 mm Range)

The manuscript states that the tag requires a 2 mm distance from the reader for operation. For real-world adoption, this limitation raises questions:

- How would such a short range be practically implemented (e.g., integration into packaging, handheld readers)? If a handheld reader is to be used, how would a large number of tags be monitored simultaneously?**
- Are there plans to improve the readout range, or is this proximity intentional (e.g., to prevent interference)?**
- A figure or schematic illustrating the envisioned use case (e.g., placement on food containers) would help readers visualize the application.**

Thank you for the comments regarding the near-field operation and the distance between the reader and the tag.

To address this comment as well as reviewer 3, we have conducted simulations to better understand the antenna pattern and its coupling with the tag. The magnetic field distribution between the reader antenna and the passive resonant structure at different separation distances is presented in Fig. S11. As can be seen, strong electromagnetic coupling is observed at 1 mm, where the H-field is highly concentrated around the resonator, indicating efficient energy transfer. At 5 mm, the field remains sufficiently strong, confirming effective coupling in short-range scenarios. However, at 20 mm, the field intensity at the resonant structure significantly decreases, resulting in weak coupling performance. These results confirm that the proposed system is optimized for short-range operation, where the reader antenna and resonant tag are positioned within a few millimeters. The field strength and coupling efficiency are strongly dependent on separation distance, which must be carefully considered in system integration.

With the presented antenna footprint, we could read the tag up to roughly 1 cm range.

Fig. S11 | The antenna and passive tag coupling theoretical visualisation at 1, 5 and 20 mm separation distances.

Therefore, the chipless tag could be detected with a handheld reader in close vicinity. To improve on the read distance, we would need to increase the antenna emitted power or make antenna footprint bigger. For the monitoring of multiple tags simultaneously, the antenna would need to be redesigned, or the operating principle changed from near-field to reflective wave. While the reflection-based method shows promise in controlled laboratory environments, it presents limitations in practical applications due to sensitivity to tag-reader positioning, environmental interference, and system integration constraints. We are currently investigating this approach further with L-slot shaped tags and the custom-made reader with the BeagleBone and NanoVNA was also designed to be compatible with reflection-based method by simply changing the antenna shape.

The following comment was added in the discussion line 454 to 456: “As presented in the manuscript, the interrogation range of less than 20 mm currently limits the realistic simultaneous detection of multiple tags. For that matter, the antenna footprint or emitted power would have to be increased.”

Finally, a schematic of the tag placement on package was already shown in Fig. 2a. At the moment, the reader needs to be placed above the tag, or the tag demands to be detached to be placed on the reader.

Reviewer #2 (Remarks to the Author):

The authors have successfully addressed all my comments and suggestions. Congratulations for the conducted work.

Thank you for the recommendations that helped us improve on the quality of the manuscript. We appreciate your congratulations and the time dedicated to reviewing this work.

Reviewer #3 (Remarks to the Author):

R2C7 (Response to comment 7): Bending Test: The authors have conducted further experiments to demonstrate the impact of bending on the performance of the antenna. However, the experiments require more depth. For example, the results presented in Figures S5 and S6 are based on only one bending cycle, resulting in a very small frequency shift. In a more realistic environment, the tag might be subjected to multiple bending cycles, potentially amplifying the frequency shift. Therefore, it is necessary to observe the frequency shift after multiple bending cycles at each radius. Consistency could be demonstrated through this experiment.

Thank you for the recommendation on testing the resonance frequency after bending. Therefore, a new experiment was conducted.

We would like to clarify that in the manuscript we do not claim that the tag is flexible. The design was not intended to be flexible. The thickness of the substrate was chosen to prevent buckling during the zinc sintering step and at the same time to have a rigid tag be used on flat surfaces, such as cardboard boxes. The tag is expected to remain in a flat configuration during its application. Nonetheless, we conducted bending tests (up to 100 bending cycles over 10- and 15-mm radius of curvature) to evaluate mechanical robustness under unintended mechanical stresses. In most cases, these curvature values are below what can be found in standard cardboard packaging applications. The resonance frequency of the tag remained detectable and shifted by less than 4 MHz after 100 cycles at 15mm radius of curvature but disappeared after 100 cycles for a smaller radius. Those new results are presented in supplementary Fig. S7 and can be found below.

Finally, as the tag was not intended to be flexible, we did not position the neutral mechanical plane nor design the materials to support bending. Designing a truly flexible version would require a full material redesign and proper mechanical engineering, which falls outside the scope of this work. Edits were added from lines 243 to 248: “Despite the tag not intended to be flexible, it maintained a detectable resonance frequency after bending over 100 cycles at a radius of curvature of 15 mm, with an observed shift in resonance of 4 MHz (Supplementary Fig. S7). A 10 mm radius of curvature leads to failure after 100 cycles with a loss of the frequency of resonance and a shift in resonance of 15 MHz after 80 cycles. In both cases, the signal amplitude got reduced overtime due to the deterioration of the zinc conductive trace and creasing of the PHBH substrate.”

Fig. S7 | Evolution of the S_{11} response of the tag with beeswax encapsulation after 100 bending cycles. a at 15 mm radius of curvature and **b** at 10 mm radius of curvature.

R2C9: Real Environment Testing: It is understood that the tag is intended for use in a real environment, making real environment testing adequate without using an anechoic chamber. While it is true that the antenna has been benchmarked, this does not apply to the proposed antenna, which has yet to be benchmarked despite its small reading range of 2mm. Therefore, a thorough investigation of the antenna, including 2D and 3D radiation patterns, is required to determine the transmission pattern of the antenna.

The antenna was designed to operate between 1 and 6 GHz. Because of its large bandwidth, it could potentially be used with various resonating tag footprints. The radiation pattern of the antenna was evaluated using simulation as well as the effect of the tag and antenna spacing. The simulation was performed using Ansys HFSS. Theoretical estimations, including input reflection coefficient (S_{11}), near-field distribution, and antenna gain, were validated within the HFSS environment under realistic boundary and excitation conditions. The antenna gain and its 3D radiation pattern are presented in newly added Fig. S9 and Fig. S10.

The antenna gain vs frequency presented in Fig. S9 shows a relatively smooth curve, indicating stable directional performance across the frequency range, making it suitable for the chosen application.

Fig. S9 | Simulated antenna gain as a function of the frequency from 1 to 6 GHz.

Fig. S10 | Antenna 3D radiation pattern at different frequencies.

Finally, the effect of distance between the antenna and tag was simulated. The magnetic field distribution between the reader antenna and the passive resonant structure at different separation distances is presented in Fig. S11. As can be seen, strong electromagnetic coupling is observed at 1 mm, where the H-field is highly concentrated around the resonator, indicating efficient energy transfer. At 5 mm, the field remains sufficiently strong, confirming effective coupling in short-range scenarios. However, at 20 mm, the field intensity at the resonant structure significantly decreases, resulting in weak coupling performance. These results confirm that the proposed system is optimized for short-range operation, where the reader antenna and resonant tag are positioned within a few millimeters. The field strength and coupling efficiency are strongly dependent on separation distance, which must be carefully considered in system integration.

Fig. S11 | The antenna and passive tag coupling theoretical visualisation at 1, 5 and 20 mm separation distances.

The following edits were added in the manuscript lines 263 to 267 to present the newly added figures S9, S10 and S11: “Simulation of the tag and antenna gain and 3D pattern indicate stable directional performance across the frequency range, making the antenna suitable for the chosen application (Supplementary **Fig. S9** and **Fig. S10**). The theoretical visualisation of the tag and antenna show that they remained coupled at 5 mm distance with a decoupling occurring at 20 mm (Supplementary **Fig. S11**).”

The method section was also updated lines 608 to 617:

“Commercially available full-wave electromagnetic solver, Ansys HFSS, was employed for the design, optimization, and theoretical performance estimation of both the reader antenna and the multi-resonant tag structure. The simulation environment enabled accurate modeling of the antenna geometry, substrate characteristics, and near-field coupling behavior critical to short-range operation. Parametric sweeps and optimization routines were used to tune the antenna

dimensions for wideband impedance matching and stable radiation performance across the multiple resonances of the tag. Theoretical estimations, including input reflection coefficient (S_{11}), near-field distribution, and antenna gain, were validated within the HFSS environment under realistic boundary and excitation conditions.”

R2C11: S11 Reduction: As shown in Figure S1, the S11 of the antenna has significantly reduced from -25dB to -8dB after one year of storage with a beeswax encapsulation. What could be the cause of this reduction, and is -8dB sufficient for the reader to pick up the tag repetitively?

We expect the reduction in amplitude to be due to the slight oxidation of the zinc overtime, which has an effect on the electrical conductivity of the printed zinc layer, as no frequency shift was observed. As we use degradable materials, encapsulating the PHBH with beeswax on both sides might help prevent this effect by protecting better the zinc film and will be considered in future developments. Nevertheless, the functionality of the 1-year-old tag was not significantly affected as the resonance could still be detected. To see if the tag can still be used, a new experiment was conducted by monitoring the aged tag with a coconut oil PCM material and analysing the induced shift after melting. The melting of the PCM of the tag was provided a shift in resonance of 34 MHz, like pristine tags. The Figure S8 was updated with this new result.

Edits: Lines 249 to 252, “While the signal amplitude got reduced after 1 year, allegedly due to the slight oxidation of the zinc trace, the aged tag remained functional and the accurate melting of a PCM for temperature-exceedance was able to be detected.”

Fig. S8 | Response of the printed chipless tag after 1 year at ambient conditions. The frequency of resonance of the tag with beeswax, after fabrication and 1 year later. The frequency of resonance of the encapsulated tags exhibits a drift of only 4 MHz after 1 year with an amplitude drop of 16 dB. After melting of the PCM, the 1-year-old tag still provided a resonance shift of 34 MHz.

Addition comment: New Table 1: Reference 56 seems to produce a large frequency shift of 600 MHz compared to the present work's 40 MHz. Is this because their antenna is more sensitive? Additionally, the term "Reproducible frequency shift" is introduced, and the present work claims to have a reproducible frequency change over other work. However, there is no data to support this claim. In addition, it would be more useful to include a comparison of S11 at their resonant frequencies and the frequency shift due to bending for all included work.

We appreciate this observation about the difference in the measured shift between our manuscript and reference 64 (previously reference 56). Reference 64 relies on the reflection wave principle, which differs fundamentally from the near-field operational mode used in our work. The observed frequency shifts are not directly comparable, as they stem from distinct working principles and significantly different design footprints. Therefore, drawing conclusions about tag sensitivity across these two approaches is not meaningful.

While the reflection-based method shows promise in controlled laboratory environments, it presents limitations in practical applications due to sensitivity to tag-reader positioning, environmental interference, and system integration constraints. However, all these factors must be carefully managed. We are currently investigating this approach further and the custom-made reader with the Beaglebone and NanoVNA was also designed to be compatible with reflection-based method by simply changing the antenna configuration.

Regarding the “reproducible frequency shift”, the terminology was replaced with “variability of frequency shift”. We note that the other studies on wireless temperature-threshold monitoring cited in the comparison table do not report data on bending or variability of the response at all, making direct comparison difficult. In contrast, we believe our work to be more complete as it reports on the drift after 1 year and includes multiple test cycles of phase change materials (PCMs) applied over various chipless tags at different conditions, along with analysis including standard deviations of the frequency shifts.

Typo in Table 1: Should the resonator be RLC, not ELC?

Thank you for noticing the typo. ELC was replaced with RLC. ELC could also be used as it stands for electric-field-coupled resonator but as you noted, it would be better to present the tags as RLC devices in the table.